# FedRAM: Federated Reweighting and Aggregation for Multi-Task Learning

**Fan Wu**[1]   **Xinyu Yan**[1]   **Jiabei Liu**[1]   **Wei Yang Bryan Lim**[1,✉]

[1]Nanyang Technological University

{fan009, xinyu020, S200131@e.ntu.edu.sg}   bryan.limwy@ntu.edu.sg

## Abstract

Federated Multi-Task Learning (FL-MTL) enables clients with heterogeneous data to collaboratively train models capable of handling multiple downstream tasks. However, FL-MTL faces key challenges, including statistical heterogeneity, task interference, and the need to balance local learning with global knowledge sharing. Traditional methods like FedAvg struggle in such settings due to the lack of explicit mechanisms to address these issues. In this paper, we propose FedRAM, a three-step framework that progressively updates two scalar hyperparameters: the task importance weight and the client aggregation coefficient. FedRAM introduces a reference-proxy-agent strategy, where the proxy model serves as an intermediate between the local reference model and the global agent model. This design reduces the need for repeated local training while preserving local performance. Extensive experiments on six real-world FL-MTL benchmarks show that FedRAM improves performance by at least $3\%$ over the most baseline on both in-domain and out-of-domain tasks, while reducing computational cost by $15\times$. These results make FedRAM a robust and practical solution for large-scale FL-MTL applications. The code is available at https://github.com/wwffvv/FedRAM.

## 1   Introduction

Federated Learning (FL) [1] is a key paradigm in distributed machine learning, enabling multiple clients to collaboratively train models while preserving data privacy. FL excels in scenarios where clients share similar domains and statistically homogeneous data distributions. However, in real-world multi-task learning (MTL) settings, significant data heterogeneity degrades global model performance and impedes effective knowledge sharing. Consequently, existing FL methods [2–4] struggle to strike a balance between in-domain performance and out-of-domain generalization in MTL contexts. To be specific, we refer to the term *domain* as the data domain accessible to clients.

A key limitation of traditional FL [1] lies in its reliance on fixed aggregation coefficients, often proportional to local sample sizes, as illustrated in Figure 1(a). While effective in homogeneous settings, this approach fails to capture task-specific variations in MTL scenarios, leading to suboptimal performance due to imbalanced client contributions. MTL settings commonly exhibit task heterogeneity and non-IID data characteristics, such as feature and label distribution skews, class imbalances, and quantity disparities [5]. To address these challenges, FL-MTL methods [6] leverage task correlations to enhance local data representations. As illustrated in Figure 1(b), incorporating similarity-based adjustments, through representations, tasks, or models, improves task alignment and local training efficacy.

Despite these advances, FL-MTL methods inadequately address imbalanced client contributions, where variations in data volume and relevance disproportionately impact the global model. This stems from their focus on in-domain performance. Recent work on model merging [7] demonstrates the

39th Conference on Neural Information Processing Systems (NeurIPS 2025).

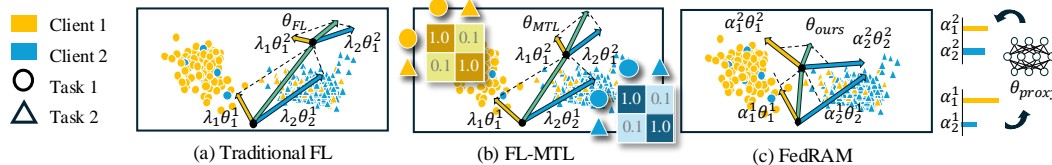

Figure 1: Comparison of aggregation strategies in federated learning. (a) Traditional FL aggregates local models using fixed coefficients $\lambda_1, \lambda_2$ (e.g., weighted by local sample sizes). (b) FL-MTL methods incorporate task correlations and data distributions to enhance local training. (c) FedRAM employs adaptive aggregation coefficients $\alpha_1, \alpha_2$, dynamically adjusted by a proxy model to balance client contributions.

efficacy of adaptive aggregation weights for task-specific models, while [8] emphasizes quantifying client contributions to ensure aggregation fairness. These insights motivate us to propose a more adaptive reweighting mechanism from three perspectives: (1) effective knowledge sharing, (2) robust generalization, and (3) accelerated convergence.

In this work, we propose FedRAM, a novel framework tailored for heterogeneous multi-task distributed environments. FedRAM employs a three-step architecture, with each model assigned a distinct role: (1) a compact local reference model $\theta_{\text{ref}}$ that captures task-specific distributions and serves as a benchmark for proxy model training; (2) a compact federated proxy model $\theta_{\text{proxy}}$ that dynamically adjusts task-specific weights and client-specific aggregation coefficients, as illustrated in Figure 1(c), and (3) a large federated agent model $\theta_{\text{agent}}$ that performs reweighted aggregation to produce the final global model. This hierarchical design enables FedRAM to adapt to in-domain variations while achieving strong out-of-domain generalization. Our contributions are summarized as follows:

- We propose FedRAM, a novel FL-MTL framework that tackles multi-task heterogeneity through a three-stage training process. Each stage employs tailored strategies aligned with the model's functional role.
- We introduce task-specific and client-specific weights as key hyperparameters in FedRAM, along with their tuning processes. These mechanisms enable fine-grained control over task prioritization and client contributions to enhance adaptability and performance.
- We conduct comprehensive experiments across diverse datasets, demonstrating that FedRAM significantly outperforms state-of-the-art methods. Our framework achieves superior accuracy in both in-domain and out-of-domain evaluations, faster convergence, and reduced computational costs, validating its effectiveness and efficiency.

## 2 Related Work

**Federated Learning.** The seminal FL approach, FedAvg [1], aggregates locally trained models into a global model by averaging client updates, offering communication efficiency and privacy benefits. However, FedAvg assumes that client data are independent and identically distributed (IID). When data distributions are non-IID, FedAvg exhibits slower convergence and reduced accuracy [9]. Subsequent works relax this IID assumption and emphasize personalization. FedProx [10] incorporates a proximal term to reduce client drift. Ditto [11] introduces client-specific regularization to personalize models while still learning a shared representation. More recently, methods like MOON [12] use a contrastive mechanism to guide local updates based on global representations. DBE [8] tackles domain biases by preserving both generic and client-specific knowledge. Despite these advances, standard FL methods predominantly focus on learning a single global model or mildly personalized models, which is often insufficient for complex multi-task scenarios.

**Federated Multi-Task Learning (FL-MTL).** Addressing the limitations of single-model FL, FL-MTL tailors models to each client's specific tasks, while still exploiting task interdependencies [6]. Early FL-MTL approaches, such as MOCHA [6] and MIFA [13], jointly learn task-specific and global parameters but may suffer from high computational costs as the number of tasks grows. More recent frameworks adopt architectural strategies to improve scalability. FedDAT [14] employs multi-modal foundation models with dual adapters, enabling efficient sharing of representations for different data modalities. FedBone [15] decouples a shared backbone from task-specific layers, balancing communication efficiency and task-specific performance.

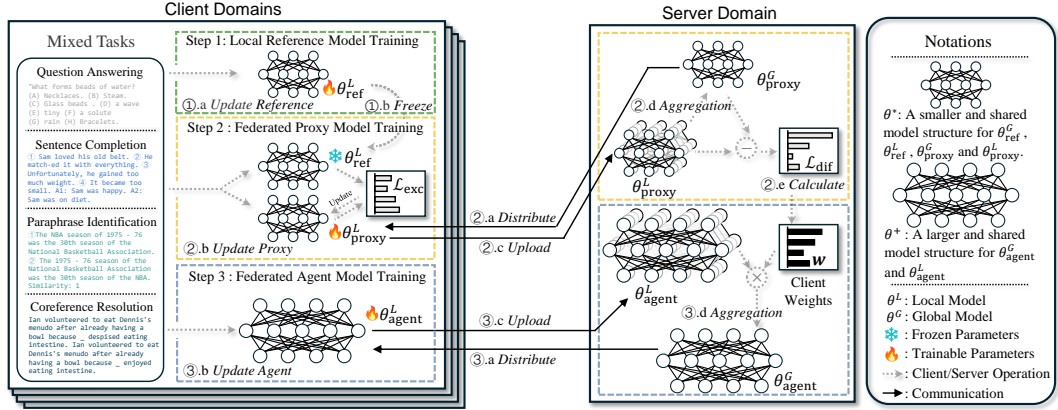

Figure 2: An overview of FedRAM framework comprising of Reference, Proxy, and Agent models.

Despite these innovations, a key challenge remains in how to fairly aggregate client contributions and mitigate interference across tasks. Naive weighted-averaging weighting (e.g., proportional to data sizes) may overlook important factors like task complexity or relevance to the global objective [9]. Several methods have proposed alternative aggregation schemes. Contribution-aware FL [16] weighs clients by measuring improvement to the global model, while [17] and [7] explore adaptive model merging to reduce parameter conflicts. Nonetheless, most existing work primarily targets either a solution to non-IID distribution or coarse-grained multi-task modeling, without an explicit reweighting module that can adapt both client- and task-level contributions in a unified federated framework.

Unlike prior work focusing solely on a single global model or static weighting heuristics, FedRAM introduces a framework that systematically adjusts task and client aggregation weights based on local performance improvements. This design alleviates inter-task conflicts and reduces computation and communication overhead by separating lightweight proxy updates from larger agent model training.

## 3 Problem Statement

Consider a scenario with total $K$ clients and each possesses a local training dataset mixed by distinct $\mathcal{T}$ tasks. We use $\mathcal{D}_k^\tau$ to present the data from the $k$-th client and $\tau$-th task ($\mathcal{D}_L$ denotes local data). The overall goal is to train an FL model that balances local task-specific distribution and adaptation to global knowledge. We consider two non-negative weights: task weights $\boldsymbol{\alpha} = [\alpha_1, ..., \alpha_\tau, ..., \alpha_\mathcal{T}]$, $\sum_{\tau=1}^{\mathcal{T}} \alpha_\tau = 1$ and client weights $\boldsymbol{w} = [w_1, ..., w_k, ..., w_K]$, $\sum_{k=1}^{K} w_k = 1$ . Building on this, the optimization goal of model $\theta$ is formulated as:

$$\min_{\theta, \boldsymbol{\alpha}, \boldsymbol{w}} \mathcal{L}(\theta; \boldsymbol{\alpha}; \boldsymbol{w}) = \sum_{k=1}^{K} w_k \sum_{\tau=1}^{\mathcal{T}} \alpha_\tau \mathcal{L}_\theta(\mathcal{D}_k^\tau). \tag{1}$$

We decouple this objective as a local loss $\mathcal{L}_{\text{local}}$ and a global loss $\mathcal{L}_{\text{global}}$:

$$\mathcal{L}_{\text{local}} = \mathcal{L}_{\theta_1}(\mathcal{D}_L) + \sum_{\tau=1}^{\mathcal{T}} \alpha_\tau (\mathcal{L}_{\theta_2}(\mathcal{D}_L^\tau) - \mathcal{L}_{\theta_1}(\mathcal{D}_L^\tau)) \tag{2}$$

We derive $\alpha_\tau$ by two steps: 1. Minimizing $\mathcal{L}_{\theta_1}(\mathcal{D}_L)$; 2. Minimizing the second term by co-optimizing $\theta_2$ and $\alpha_\tau$. We assign $\theta_1$ and $\theta_2$ with the same model architecture.

$$\mathcal{L}_{\text{global}} = \sum_{k=1}^{K} w_k \mathcal{L}_{\text{local}} \tag{3}$$

Our proposed framework follows a three-step sequential solution: $\boldsymbol{\alpha}$, $\boldsymbol{w}$ and $\theta_3$, respectively. In our implementation, we refer to the three distinct models as $\theta_{\text{ref}}$, $\theta_{\text{proxy}}$, and $\theta_{\text{agent}}$.

## 4 Proposed Method

As illustrated in Figure 2, FedRAM employs distinct notations: $\theta^G$ and $\theta^L$ differentiate models deployed on the server (**G**lobal model) and client devices (**L**ocal models), respectively; $\theta^*$ and $\theta^+$

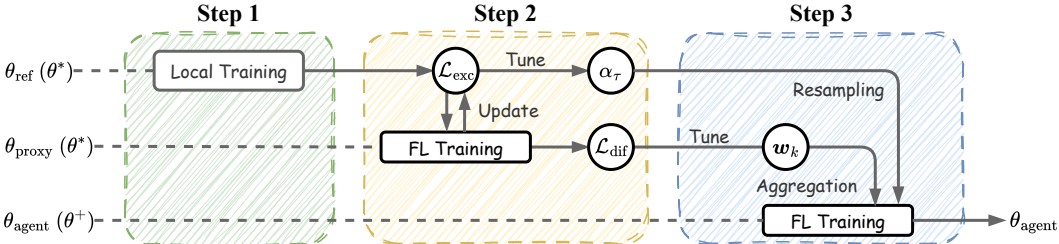

Figure 3: A functional overview of the three models in FedRAM.

distinguish between smaller and larger model architecture. Specifically, FedRAM consists of three main functional models (Figure 3), summarized as follows:

- **Reference.** With previously training in the local client's domain in **Step 1**, the frozen $\theta_{\mathrm{ref}}$ (**Step 2**) is kept local and then used to calculate the excess loss $\mathcal{L}_{\mathrm{exc}}$, which helps train $\theta_{\mathrm{proxy}}$ and adjust the task weights $\alpha_\tau$.

- **Proxy.** By applying a smaller model architecture $\theta^*$, the FL-trained $\theta_{\mathrm{proxy}}$ learns prior knowledge of task heterogeneity at a lower cost. In **Step 3**, $\theta_{\mathrm{proxy}}$ tunes the client weights $\boldsymbol{w}_k$.

- **Agent.** At the final stage of FedRAM, larger model $\theta_{\mathrm{agent}}$ is trained in FL for final evaluation. The training process incorporates the tuned parameters $\alpha_\tau$ (for resampling local data) and $\boldsymbol{w}_k$ (for weighted aggregation), which are obtained through $\theta_{\mathrm{ref}}$ and $\theta_{\mathrm{proxy}}$.

According to the Algorithm 1 for FedRAM, we illustrate each step in the following sections.

## 4.1 Step 1: Local Reference Model Training

FedRAM benefits from local reference model training in two ways: 1. A set of locally adapted reference models $\theta_{\mathrm{ref}}^L$ enables the adaptation process to local multi-task distributions; 2. Applying a lightweight model architecture lowers the computation cost. The key steps are as follows:

- **Model Initialization:** Clients receive a globally shared model structure $\theta^*$.

- **Local Optimization Objective:** The reference models are optimized to gain a better representation of local distributions:

$$\min_{\theta_{\mathrm{ref}}^L} \mathcal{L}(\theta_{\mathrm{ref}}^L) = \frac{1}{|\mathcal{D}_L|} ||f_{\theta_{\mathrm{ref}}^L}(x) - y||^2. \tag{4}$$

where $(x, y)$ denotes the data and label in local dataset $\mathcal{D}_L$. $|\mathcal{D}_L|$ is the dataset size.

- **Local Updates:** Clients independently and locally update the parameters of $\theta^*$, thus deriving the reference models $\theta_{\mathrm{ref}}^L$.

$$\theta_{\mathrm{ref}}^L \leftarrow \theta_{\mathrm{ref}}^L - \beta \nabla_{\theta_{\mathrm{ref}}^L} \mathcal{L}(\mathcal{D}_L; \theta_{\mathrm{ref}}^L) \tag{5}$$

where $\beta$ represents the learning rate, $\mathcal{L}$ represents the loss function.

- **Parameters Freezing:** The well-trained reference models $\theta_{\mathrm{ref}}^L$ freeze their parameters and are kept within the local domain. Specifically, in Section 4.2, the inference outputs of $\theta_{\mathrm{ref}}^L$ are utilized to compute excess losses for training the proxy models $\theta_{\mathrm{proxy}}^L$.

Once each client has updated their reference model and reported to the server, the server would embark on the **Step 2** proxy model training process.

## 4.2 Step 2: Federated Proxy Model Training

In this subsection, we introduce the excess loss updates in the *Local Training* process and the task-weights-based aggregation scheme in the *Global Aggregation* process.

### 4.2.1 Local Training of Proxy Model

In a local training round, the proxy model employs group distributionally robust optimization [18] to fine-tune the model parameters $\theta^L_{\text{proxy}}$. The loss update process consists of the following steps:

- **Initialization:** To align with reference models, the proxy models $\theta^G_{\text{proxy}}$ and $\theta^L_{\text{proxy}}$ are randomly initialized with $\theta^*$ for lightweight training.

- **Task-wise Loss Computation:** To better measure the contributions by distinct tasks, task-wise datasets $\mathcal{D}^\tau_k \in \mathcal{D}_k$ are divided within the domain of client $k$. The split of different tasks aligns with the task weights $\alpha_\tau, (\tau = 1, 2, ..., \mathcal{T})$. The task-wise loss is denoted as $\ell^\tau = \mathcal{L}(\mathcal{D}^\tau_L; \theta^L_{\text{proxy}})$.

- **Excess Loss Calculation:** Following the notation of task-wise losses $\ell^\tau$, the excess loss calculation involves the inference loss $\ell^\tau_{\text{ref}}$ of frozen reference model and $\ell^\tau_{\text{proxy}}$ of the updating proxy model. To quantify the potential improvement between the federated proxy performance and the reference headroom [19], the task-wise excess loss is calculated as:

$$\ell^\tau_{\text{exc}} = \max \left\{ \frac{1}{|\mathcal{D}^\tau_L|}(\ell^\tau_{\text{proxy}} - \ell^\tau_{\text{ref}}), 0 \right\} \tag{6}$$

  where $|\mathcal{D}^\tau_L|$ is the dataset size of $\mathcal{D}^\tau_L$. Note that positive values mean that the proxy model represents the local distributions better than the reference model.

- **Dynamic Task Weight Adjustment:** Task-specific weights $\alpha$ ($\alpha \in \mathbb{R}^{n \times 1}$) are iteratively adjusted:

$$\alpha_\tau \leftarrow \alpha_\tau e^{\eta_\tau \ell^\tau_{\text{exc}}} \tag{7}$$

  where $\eta_\tau \in \eta_{\text{task}}$ is an exponential scaling factor for task $\tau$, controlling the scaling span. For a hyperparameter study of the exponential scaling factor $\eta_\tau$, please refer to Appendix B.3. A smooth parameter $s$ is introduced to balance $\alpha^-$ and the dynamically calculated $\alpha^+$: $\alpha = s\alpha^- + (1 - s)\alpha^+$. Besides, normalization is employed to bound $\alpha$ and ensure $\sum_{\tau=1}^{\mathcal{T}} \alpha = 1$.

- **Local Optimization Objective:** The proxy model's local optimization aims to minimize the combined weighted excess losses across tasks:

$$\min_{\theta^L_{\text{proxy}}} \mathcal{L}(\theta^L_{\text{proxy}}, \alpha) = \sum_{\tau=1}^{\mathcal{T}} \alpha_\tau \ell^\tau_{\text{exc}}. \tag{8}$$

  This objective ensures that local training addresses the specifics of each task.

### 4.2.2 Global Aggregation of Proxy Models

FL aggregation facilitates global knowledge sharing related to task-specific adjustments. FedAvg is adopted as a baseline in the aggregation of proxy models:

$$\theta^G_{\text{proxy}}(t) \leftarrow \theta^G_{\text{proxy}}(t - 1) + \sum_{k=1}^{K} \frac{|\mathcal{D}_k|}{\sum |\mathcal{D}_k|} \Delta \theta^k_{\text{proxy}}(t), \tag{9}$$

where $t$ denotes the $t$-th communication round, and $k$ represents the $k$-th client. In the following sections, the symbol $\theta^k$ instead of $\theta^L$ is used to explicitly denote the $k$-th local model.

### 4.3 Step 3: Federated Agent Model Training

*Agent* models $\theta^L_{\text{agent}}$ and $\theta^G_{\text{agent}}$ act as the decision-maker and perform a primary evaluation subject to FedRAM framework.

**Resampling Client Dataset.** By utilizing the task weights generated in Step 2, at the start of each communication round, resampling each client domain ensures the heterogeneity in the MTL setting can be contributed proportionally to the global learning objective.

**Optimizing Client Weights.** FedRAM adjusts the merging weights $w_k$ for each agent model based on the differential loss. Similar to Equation 6 and 7, the differential loss $\ell_{\text{dif}}$ is defined as below:

$$\ell_{\text{dif}} = \frac{1}{|\tilde{\mathcal{D}}^\tau_L|}(\ell^L_{\text{proxy}} - \ell^G_{\text{proxy}}) \tag{10}$$

and the client merging weights are formulated by:

$$w_k \leftarrow w_k e^{\eta_k \ell_{\text{dif}}^k}, \tag{11}$$

where $\eta_k$ is an exponential scaling factor. Here, a smoothing parameter and normalization can be introduced as in Equation 7. A hyperparameter study of $\eta_k$ is provided in Appendix B.3.

**Local Optimization Objective:** The optimization aims to minimize the local losses:

$$\min_{\theta_{\text{agent}}^L} \mathcal{L}(\mathcal{D}_L; \theta_{\text{agent}}^L). \tag{12}$$

**Global Aggregation and Model Finalization.** After optimizing the aggregation weight, the global model aggregation is performed using reweighted client weights:

$$\theta_{\text{agent}}^G(t) \leftarrow \theta_{\text{agent}}^G(t-1) + \sum_{k=1}^{K} w_k \Delta\theta_{\text{agent}}^k(t). \tag{13}$$

where the $\theta_{\text{agent}}^k$ instead of $\theta_{\text{agent}}^L$ is used to provide a more fine-grained representation of client $k$. To further improve FedRAM, various aggregation methods can be used as an alternative combination. We provide the convergence proof in Appendix A.

---

**Algorithm 1:** FedRAM

---

**Input:** Base models $\theta^*, \theta^+$ (with initialization $\theta_{\text{ref}}^L, \theta_{\text{ref}}^G, \theta_{\text{proxy}}^L, \theta_{\text{proxy}}^G \leftarrow \theta^*; \theta_{\text{agent}}^L, \theta_{\text{agent}}^G \leftarrow \theta^+$),
    number of clients $K$, exponential scaling factors $\eta_{\text{task}}$ and $\eta_{\text{client}}$, smoothing parameter $s$,
    client $k$ training data $\mathcal{D}_k$, learning rate $\beta$
**Output:** Global agent model $\theta_{\text{agent}}^G$

1  **STEP 1: Train Reference Model**
2  **for** *client $k = 1, 2, ..., K$* **do**
3  $\quad \lfloor \; \theta_{\text{ref}}^k \leftarrow \theta_{\text{ref}}^k - \beta \nabla_{\theta_{\text{ref}}^k} \mathcal{L}(\mathcal{D}_k; \theta_{\text{ref}}^k)$ $\qquad\qquad\qquad\qquad\qquad$ ▷ Local Training

4  **STEP 2: Train Proxy Model**
5  **for** *Communication round $t = 1, 2, ..., T$* **do**
6  $\quad$ **for** *client $k = 1, 2, ..., K$* **do**
7  $\quad\quad$ $\theta_{\text{proxy}}^k(t) \leftarrow \theta_{\text{proxy}}^G(t-1); \quad \alpha(t) \leftarrow \alpha(t-1)$ $\qquad$ ▷ Update Last Round Parameters
8  $\quad\quad$ **for** *task $\tau = 1, 2, ..., \mathcal{T}$* **do**
9  $\quad\quad\quad \lfloor \; \ell_{\text{exc}}^{\tau} \leftarrow \max\left\{ \left( \nabla_{\theta_{\text{proxy}}^k} \mathcal{L}(\mathcal{D}_k^{\tau}; \theta_{\text{proxy}}^k) - \nabla_{\theta_{\text{ref}}^k} \mathcal{L}(\mathcal{D}_k^{\tau}; \theta_{\text{ref}}^k) \right) / |\mathcal{D}_k^{\tau}|, 0 \right\}$
10 $\quad\quad$ $\alpha(t) \leftarrow \alpha(t) \cdot e^{\eta_{\text{task}} \mathcal{L}_{\text{exc}}}$ by $\mathcal{L}_{\text{exc}} = [\ell_{\text{exc}}^1, \ell_{\text{exc}}^2, \ldots, \ell_{\text{exc}}^{\mathcal{T}}]$ $\qquad\qquad$ ▷ Update $\alpha$
11 $\quad\quad$ $\theta_{\text{proxy}}^k(t) \leftarrow \theta_{\text{proxy}}^k(t) - \beta \cdot \alpha(t) \cdot \mathcal{L}_{\text{exc}}$ $\qquad\qquad\qquad$ ▷ Update Local Proxy Model
12 $\quad\quad$ **Last Epoch:** $\ell_{\text{dif}}^k \leftarrow \left( \nabla_{\theta_{\text{proxy}}^k} \mathcal{L}(\mathcal{D}_k) - \nabla_{\theta_{\text{proxy}}^G} \mathcal{L}(\mathcal{D}_k) \right) / |\mathcal{D}_k|$
13 $\quad$ $\mathcal{L}_{\text{exc}} = [\ell_{\text{exc}}^1, \ell_{\text{exc}}^2, \ldots, \ell_{\text{exc}}^K]$
14 $\quad$ $\boldsymbol{w}(t) \leftarrow \boldsymbol{w}(t-1) \cdot e^{\eta_{\text{client}} \mathcal{L}_{\text{exc}}}$ $\qquad\qquad\qquad\qquad\qquad$ ▷ Server Update $\boldsymbol{w}$
15 $\quad$ $\theta_{\text{proxy}}^G(t) \leftarrow \theta_{\text{proxy}}^G(t-1) + \sum_k \Delta\theta_{\text{proxy}}^k(t)$ $\qquad\qquad\qquad$ ▷ Server Aggregation

16 **STEP 3: Train Agent Model**
17 **for** *Communication round $t = 1, 2, ..., T$* **do**
18 $\quad$ **for** *client $k = 1, 2, ..., K$* **do**
19 $\quad\quad$ $\theta_{\text{agent}}^k(t) \leftarrow \theta_{\text{agent}}^G(t-1)$ $\qquad\qquad$ ▷Resample the task datasets by size and $\alpha$
20 $\quad\quad$ $\theta_{\text{agent}}^k(t) \leftarrow \theta_{\text{agent}}^k(t) - \beta \cdot \nabla_{\theta_{\text{agent}}^k} \mathcal{L}(\mathcal{D}_k; \theta_{\text{agent}}^k)$ $\qquad$ ▷ Update Local Agent Model
21 $\quad$ $\theta_{\text{agent}}^G(t) \leftarrow \theta_{\text{agent}}^G(t-1) + \sum_k w_k \Delta\theta_{\text{agent}}^k(t)$ $\qquad\qquad$ ▷ Server Aggregation

---

## 5 Experiment

### 5.1 Experiment Settings

**Datasets.** Following the work in [17] and [7], we conduct experiments based on four diverse categories of NLP datasets, each corresponding to specific task types, including: (1) Question

Answering: *QASC* [20], *WikiQA* [21], and *QuaRTz* [22]; (2) Paraphrase Identification: *PAWS* [23]; (3) Coreference Resolution: *Winogrande* [24] and *WSC* [25]; (4) Sentence Completion: *Story Cloze* [26]. In constructing mixed local datasets, we especially focus on two settings as follows.

- *Setting 1*: The number of clients $K$ larger than the number of tasks $\mathcal{T}$ ($K \geq \mathcal{T}$). We primarily consider $K = 10$ and $\mathcal{T} = 7$ in our main text. Thorough statistics of data distributions are provided in Figure 5 in Appendix B.

- *Setting 2*: An extreme heterogeneous case with each client assigned one distinct task. We provide a case study under Setting 2 in Appendix B.

**Training Setup.** In the experiments, we employ a global LoRA configuration to fine-tune the parameters. We adopt T5-small model as $\theta^*$ and T5-base model as $\theta^+$. We assume equal values for exponential scaling factors ($\eta_\tau$ and $\eta_k$) and smoothing parameter $s$. Our simulations are conducted on a cloud instance, equipped with 8 NVIDIA A10 GPUs (24 GiB of memory per GPU), 128 vCPUs (Intel Xeon Platinum 8369B), and 512 GB RAM. For the three-stage training, we employ cross-entropy loss with an Adam optimizer, setting the learning rate $\beta$ to $1 \times 10^{-3}$. We set the maximum global rounds to 50. For simplicity, we assume that all clients can participate in every communication round. More details about our experiment implementation and baselines can be found in Table 4 in Appendix B.

**Evaluation Metrics.** We evaluate model performance using both global and local held-out validation data. Specifically, we consider both the In-Domain (ID) and Out-of-Domain (OOD) evaluation strategies to assess the generalization capabilities of the model:

- *In-Domain Evaluation*: Evaluating the model on tasks that are locally accessible to individual clients. ID evaluation helps measure how well the model adapts to local data during training.

- *Out-of-Domain Evaluation*: Assessing the model on tasks that were introduced by other clients and are locally inaccessible. OOD Acc reflects the model's robustness and ability to generalize.

$$\text{ID Acc} = \frac{\sum_{\tau=1}^{\mathcal{T}} \text{Acc}_\tau^{(\tau)} |\mathcal{D}_L^\tau|}{\sum_{\tau=1}^{\mathcal{T}} |\mathcal{D}_L^\tau|}, \qquad \text{OOD Acc} = \frac{\sum_{\tau=1}^{\mathcal{T}} \sum_{\hat{\tau} \neq \tau} \text{Acc}_\tau^{(\hat{\tau})} |\mathcal{D}^{\hat{\tau}}|}{\sum_{\tau=1}^{\mathcal{T}} \sum_{\hat{\tau} \neq \tau} |\mathcal{D}^{\hat{\tau}}|}. \quad (14)$$

where $\tau$ and $\hat{\tau}$ denotes task $\tau$ and $\hat{\tau}$, $\text{Acc}_\tau^{(\tau)}$ denotes the accuracy for a client trained on task $\tau$ while tested on task $\tau$, $|\mathcal{D}_L^\tau|$ is the dataset of task $\tau$ and $|\mathcal{D}_L^\tau|$ is the sample size of task $\tau$.

## 5.2 Results and Discussion

### 5.2.1 Main Results

Table 1: Comparison of FL methods across diverse tasks using global, local, and in-domain/out-of-domain (ID/OOD) evaluation metrics. FedRAM achieves superior F1-scores in global and local validations, as well as competitive ID/OOD performance. Scores in **bold** denote the best performance, and underlined scores indicate the second-best.

| Methods | Tasks | | | | | | | Global F1-Score | Local F1-Score / Bottom Decile | ID / OOD Evaluation |
|---|---|---|---|---|---|---|---|---|---|---|
| | PAWS | WSC | Wino Grande | QASC | Qua-RTz | Story Cloze | Wiki QA | | | |
| FedAvg [1] | 83.36 | **77.90** | 75.99 | 32.44 | 78.30 | 82.20 | 79.78 | 72.71 | 76.34 / 54.79 | 71.68 / 75.66 |
| FedProx [10] | 78.45 | 73.39 | 80.29 | 28.87 | 78.83 | 76.77 | 66.35 | 68.83 | 73.73 / 55.45 | 70.30 / 72.03 |
| Ditto [11] | 83.25 | 77.69 | 77.65 | 30.04 | 77.97 | 80.28 | 79.79 | 72.26 | 77.09 / 62.37 | 70.83 / 71.75 |
| FedRep [27] | 74.56 | 63.55 | 76.91 | 34.58 | 64.71 | 82.45 | 66.35 | 66.95 | 76.89 / 66.89 | 65.48 / 55.79 |
| MOON [12] | 74.45 | 73.39 | 78.88 | 31.91 | 79.21 | 78.24 | 79.79 | 70.31 | 72.71 / 56.30 | 69.70 / 71.66 |
| DBE [8] | 81.22 | 76.44 | 76.25 | 30.82 | 78.40 | 81.56 | 79.79 | 72.08 | 75.56 / 58.53 | 68.09 / 73.16 |
| FedMTL [28] | **89.04** | 74.90 | 78.77 | 32.10 | 66.05 | **84.54** | 79.73 | 74.12 | 78.88 / 62.32 | **76.01** / 75.64 |
| FedBone [15] | 85.31 | 75.50 | 78.32 | 35.48 | 55.24 | 81.24 | 79.79 | 71.19 | 71.85 / 61.60 | 67.97 / 70.39 |
| FedRAM (Ours) | 81.11 | 77.69 | **80.78** | **36.15** | **80.36** | 77.62 | **79.79** | **75.94** | **79.62** / **73.21** | 72.89 / **76.32** |

Table 1 presents a comprehensive evaluation of FedRAM against SOTA FL methods across diverse tasks and metrics. FedRAM consistently outperforms baselines in global and local F1-scores, particularly in the bottom decile, while achieving competitive ID and OOD performance. Key observations include: (i) FedRAM achieves the optimal global and task-wise performance, with a global F1-score of 75.94, achieving the highest performance in 4 out of 7 tasks. (ii) On the QASC

task, all methods exhibit a drop in F1-score, while FedRAM maintains relatively high performance. This decline is attributed to the inherent label imbalance in the multiple-choice answer selection task. (iii) In terms of local evaluation, FedRAM achieves the highest average score of 79.62 among clients, with a bottom decile score of 73.21. Notably, the bottom decile typically occurs in the *QASC-dominated* client (possessing the largest proportion of QASC data), where severe label imbalance poses a significant challenge. While FedMTL and FedRep achieve competitive overall performance across clients, they fail to directly enhance the performance of the *QASC-dominated* client. (iv) For ID and OOD evaluation, FedRAM achieves scores of 72.89 (*2nd*) / 76.32 (*1st*).

Compared to weight adjustment strategies such as DBE and non-adjustment baselines (FedAvg, FedProx), FedRAM demonstrates a significant improvement of at least 3% across multiple metrics. This improvement validates FedRAM's improvement through adjusting task sample rates and client aggregation weights. FedRAM exhibits competitive performance in comparison to other task-correlation considered methods (including MOON, FedMTL, and FedBone). FedRAM offers a well-balanced performance across both ID and OOD evaluations, though FedMTL shows better ID performance (which is greatly skewed by the biased performance of PAWS and Story Cloze).

### 5.2.2 Convergence Analysis

Figure 4 demonstrates that integrating FedRAM at **Step 3** with established FL methods such as FedAvg, FedMTL, and FedBone, consistently accelerates convergence compared to these methods alone. As depicted in Figure 4(a), FedRAM yields a modest yet significant improvement in loss convergence when combined with FedAvg, though FedAvg achieves a slightly lower final accuracy. Notably, FedRAM improves the global and local F1-score by 3% over FedAvg (Table 1) and reduces the rounds to convergence by 15% (Table 5). Figure 4(b) highlights the substantial performance boost from integrating FedRAM with FedMTL, achieving faster and more efficient loss reduction compared to FedMTL alone. Similarly, Figure 4(c) shows that combining FedRAM with FedBone markedly enhances loss reduction. These results underscore FedRAM's effectiveness in dynamically adapting weights and learning rates to local data characteristics and task-specific requirements.

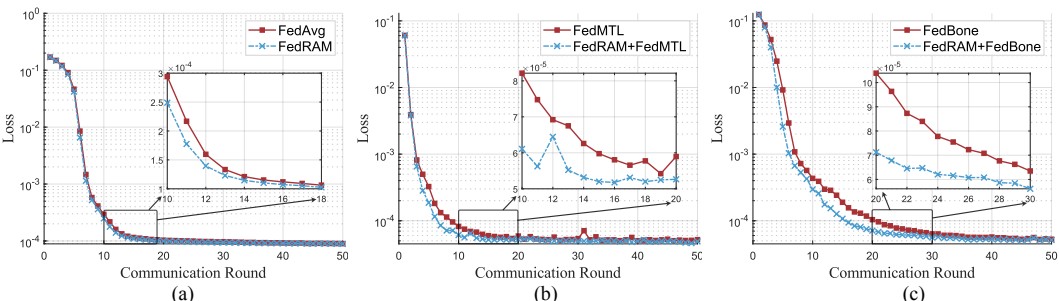

Figure 4: Convergence analysis of FedRAM integrated at **Step 3** with (a) FedAvg, (b) FedMTL, and (c) FedBone, compared to baseline methods.

### 5.2.3 Computational and Communication Costs

Computational cost, defined as the total FL training time, depends on the number of communication rounds $R$ and local computational complexity, while communication cost arises from the repeated transmission of a considerable number of model parameters during training. Table 2, DBE incurs the highest computational overhead (34,173s) due to its dual-module training mechanism. In contrast, FedRAM achieves a competitive computational cost (12,060s) through the following optimizations:

**Decoupled Training:** Lightweight reference and proxy models ($\theta^*$) requiring only 10-15% of total computation resources. As detailed in Table 5 of Appendix B.4, the agent model $\theta_{\text{agent}}$ consistently accounts for 85.08%-88.02% of total computation time across varying client scales ($N = 10$ to $50$).

**Reweighted Aggregation Coefficients:** FedRAM dynamically adjusts client update weights and learning rates during aggregation, optimizing convergence by prioritizing updates aligned with local data characteristics and task-specific requirements, as detailed in Section 5.2.2. Consequently, FedRAM reduces convergence rounds by 15% relative to FedAvg. While FedGF and FedMTL converge in fewer rounds, their computational costs are significantly higher than FedRAM's 12,060s. In contrast, FedRAM achieves a superior balance of computational efficiency and model accuracy.

Table 2: Comparison of computational cost, convergence rounds, and communication cost across FL methods. $R$ denotes the convergence rounds for $\theta^+$, $S$ denotes the convergence rounds for $\theta^*$.

| Algorithm | Computational Cost (s) | Convergence Rounds | Communication Cost |
|---|---|---|---|
| FedAvg [1] | 13774 | 21 | $R \times K \times 2\theta^+$ |
| FedProx [10] | 24195 | 25 | $R \times K \times 2\theta^+$ |
| Ditto [11] | 27782 | 27 | $R \times K \times 2\theta^+$ |
| DBE [8] | 34173 | 27 | $R \times K \times 2\theta^+$ |
| FedMTL [28] | 24502 | 17 | $R \times K \times 2\theta^+$ |
| FedBone [15] | 40989 | 30 | $R \times K \times 2\theta^+$ |
| FedGF [29] | 15596 | **16** | $R \times K \times 2\theta^+$ |
| FedRAM (Ours) | **12060** | 18 | $S \times K \times 2\theta^* + R \times K \times 2\theta^+$ |

#### 5.2.4 Ablation Studies

We systematically evaluate the contributions of the functional models in FedRAM: $\theta_{\text{ref}}$, $\theta_{\text{proxy}}$, and $\theta_{\text{agent}}$. Table 3 presents the performance of various ablated configurations, measured by Global and Local F1-Scores. Note that to maintain the three-step pipeline integrity, ablated components are replaced with randomly initialized surrogates, and evaluations are conducted

Table 3: Ablation study of FedRAM main models.

| Exp. | Method | Eval. | Global F1 | Local F1 |
|---|---|---|---|---|
| 1 | w/o $\theta_{\text{ref}}$, $\theta_{\text{proxy}}$ | $\theta_{\text{agent}}$ | 72.71 | 76.34 |
| 2 | w/o $\theta_{\text{ref}}$ | $\theta_{\text{agent}}$ | 71.17 | 73.58 |
| 3 | w/o $\theta_{\text{proxy}}$ | $\theta_{\text{ref}}$ | 70.52 | 72.54 |
| 4 | w/o $\theta_{\text{agent}}$ | $\theta_{\text{proxy}}$ | 2.69 | 2.51 |
| 5 | FedRAM | $\theta_{\text{agent}}$ | 75.94 | 79.62 |

across these different model configurations. In our ablation study, we assess each model by comparing the complete FedRAM (Exp. 5) with configurations where specific models are excluded (Exp. 1-4).

(1) Reference Model: When assessing the only reference model (Exp. 3), both global and local metrics exhibit a drop, demonstrating its limitations due to smaller parameter size. Comparing Exp. 2 and 5, excluding $\theta_{\text{ref}}$ (Exp. 2) results in a notable performance decline relative to the baseline, underscoring the critical role of local training in addressing non-IID data challenges.

(2) Proxy Model: Combining $\theta_{\text{ref}}$ and $\theta_{\text{proxy}}$ without $\theta_{\text{agent}}$ (Exp. 4) leads to a significant performance drop, as $\theta_{\text{proxy}}$ cannot independently update its parameters on local datasets. This behavior stems from $\theta_{\text{proxy}}$'s design, which aligns closely with $\theta_{\text{ref}}$ as a locally trained benchmark. The proxy model first solves for model-agnostic hyperparameters $\alpha$ and $w$ to avoid higher computational cost imposed by correlated minimax training across hyperparameter tuning and agent model training. Comparing Exp. 2 (without $\theta_{\text{proxy}}$, F1: 71.17) and Exp. 5 (F1: 75.94), the proxy model's contribution to optimizing aggregation coefficients is evident.

(3) Agent Model: Exp. 1 and 2 reveal the performance contribution of $\theta_{\text{agent}}$, which serves as the baseline model. Although there is a performance drop compared to Exp. 5, we can conclude that building upon the agent model's baseline performance, the reference and proxy models contribute by fine-tuning aggregation coefficients.

#### 5.2.5 Extensions

**Hyperparameter Sensitivity.** Our empirical analysis reveals that the optimization of task and client weights is highly sensitive to two key hyperparameters, $\eta_{\text{task}}$ and $\eta_{\text{client}}$. A comprehensive sensitivity analysis of these hyperparameters is provided in Appendix B.3, revealing their impact.

**Model Scalability.** We leverage a compact proxy model to optimize client weights, which are then directly applied to enhance agent training at a significantly larger scale (up to 15×). The choice of the reference/proxy model, when the agent model is fixed, critically influences the client weights derived by FedRAM. To investigate the minimal viable size of the reference/proxy model, we introduce an Agent-to-Proxy (A/P) ratio, with detailed evaluations presented in Appendix B.4.

**Client Scalability.** In Section 5.2.1, we evaluate FedRAM in *Setting 1* with a client number $K$ set to be 10. To assess scalability in real-world scenarios with larger client populations, we extend the analysis to client counts of 20 and 50. These experiments, detailed in Appendix B.5, confirm FedRAM's robustness and scalability across varying client numbers.

**Task Scalability.** The heterogeneity in tasks and data distributions can significantly influence the task and client weights adjusted by FedRAM. For instance, task weights enhance performance by

mitigating proxy excess loss (Section 4.2) and enabling resampling of data distributions for agent model training (Section 4.3). Further analyses, including task size variations (e.g., **Heterogeneous Tasks** under *Setting 2*) and alternative task types (e.g., **Vision Tasks**), are discussed in Appendix B.6.

# 6    Conclusion

In this paper, we propose FedRAM, a novel FL-MTL framework. To address task heterogeneity, we decouple the primary learning objective into phased sub-objectives, leveraging three distinct functional models with tailored learning strategies. We validate FedRAM's effectiveness through comprehensive metrics, including ID and OOD performance, convergence, computational cost, and ablation studies. Additionally, we provide convergence analysis and supplementary experiments in Appendices A and B, respectively.

# 7    Acknowledgement

This research is supported by the NTU startup grant and the RIE2025 Industry Alignment Fund – Industry Collaboration Projects (IAF-ICP) (Award I2301E0026), administered by A*STAR, as well as supported by Alibaba Group and NTU Singapore through Alibaba-NTU Global e-Sustainability CorpLab (ANGEL). This research / project is supported by A*STAR under its Japan-Singapore Joint Call: JST-A*STAR 2024 (Project ID: R24I6IR139). Any opinions, findings and conclusions or recommendations expressed in this material are those of the author(s) and do not reflect the views of A*STAR.

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

# A Convergence Analysis

## A.1 Theorem 1: Convergence of Standard FedAvg

We begin by recalling the standard convergence analysis for FedAvg under the standard assumptions of the main text.

**Theorem A.1** (Convergence of FedAvg). *Let $\mathcal{L}_k(\theta)$ be the local loss function for each client $k$, satisfying the L-smoothness, bounded variance, and bounded gradient norm assumptions. Under these assumptions, the global model converges to a stationary point $\theta^*$ with the following bound on the convergence rate after $T$ rounds:*

$$\mathbb{E}[\mathcal{L}(\theta^T) - \mathcal{L}(\theta^*)] \leq \frac{C}{T},$$

*where $C$ is a constant that depends on the initial conditions and the smoothness parameter $L$.*

**Proof of Theorem 1.** The proof follows from standard results in FL, as in [10]. The core idea is to bind the decrease in the global loss function at each iteration by using the smoothness of the local loss functions and the bounded variance of the stochastic gradients. Specifically, we use the descent lemma:

$$\mathcal{L}(\theta^{t+1}) \leq \mathcal{L}(\theta^t) + \langle \nabla \mathcal{L}(\theta^t), \theta^{t+1} - \theta^t \rangle + \frac{L}{2} \|\theta^{t+1} - \theta^t\|^2,$$

and show that the expected decrease is proportional to $1/T$, leading to the convergence rate of $\mathcal{O}(1/T)$. For more details, see [10].

## A.2 Theorem 2: Convergence of FedRAM with Dynamic Weights and Proxy Model

We now provide detailed proof of the convergence result for FedRAM, incorporating dynamic weight adjustments mediated by the proxy model.

**Theorem A.2** (Convergence with Adjusted Aggregation Weights in FedRAM). *Let $\mathcal{L}_k(\theta)$ be the local loss function for each client $k$, satisfying the L-smoothness and bounded variance assumptions. Then, with weight adjustments $\alpha_k$ moderated by a proxy model based on differential loss, the global model in FedRAM converges to a stationary point $\theta^*$, with the following bound on the convergence rate after $T$ rounds:*

$$\mathbb{E}[\mathcal{L}(\theta^T) - \mathcal{L}(\theta^*)] \leq \frac{\tilde{C}}{T} + \frac{\tilde{\sigma}^2}{P} \left( \frac{1}{K} \sum_{k=1}^{K} \frac{1}{n_k} \right) + \mathcal{O}(\epsilon),$$

*where $\tilde{C}$ is a constant reflecting the weight adjustments and $\tilde{\sigma}^2$ represents the reduced variance in gradients influenced by the proxy model, and $\mathcal{O}(\epsilon)$ captures the error due to task heterogeneity across clients.*

**Proof of Theorem 2.** The proof for the convergence with adjusted weights now includes the influence of the proxy model, which acts as a mediator to adjust the weights based on client-specific excess loss and task complexity.

**Step 1: Descent Lemma with Proxy Model Involvement.** By the L-smoothness assumption and incorporating the proxy model's adjustments, we apply the descent lemma for any iterate $\theta^t$:

$$\mathcal{L}(\theta^{t+1}) \leq \mathcal{L}(\theta^t) + \langle \nabla \mathcal{L}(\theta^t), \theta^{t+1} - \theta^t \rangle + \frac{L}{2} \|\theta^{t+1} - \theta^t\|^2.$$

The global model update rule now reflects the proxy model's influence:

$$\theta^{t+1} = \theta^t + \sum_{k=1}^{K} \boldsymbol{w}_k \Delta \theta_k^t,$$

where $\boldsymbol{w}_k$ are the weights adjusted by the proxy model based on the assessment of each client's needs and contributions.

**Step 2: Variance Reduction via Aggregation Coefficients Enhanced by Proxy Model.** The proxy model enhances the weight adjustments $\boldsymbol{w}_k$, leading to an optimized reduction in the variance of the aggregated gradient updates, thus improving the stability and effectiveness of the learning process. The adjusted variance is now denoted by $\tilde{\sigma}^2$, which accounts for the proxy's role in minimizing discrepancies in client updates.

**Variance Decomposition of Federated Aggregation.** Let the local client gradient be $g_L^t = \nabla\mathcal{L}_k(\theta^t)$, and the global aggregated gradient is:

$$g_{\mathrm{G}}^t = \sum_{k=1}^K \boldsymbol{w}_k \nabla\mathcal{L}_k(\theta^t),$$

where the adjusted weights satisfy $\sum_{k=1}^K \boldsymbol{w}_k = 1$. The variance decomposes as the combination of the inter-client variance and cross-client variance:

$$\mathbb{E}\left\|g_{\mathrm{G}}^t - \mathbb{E}[g_{\mathrm{G}}^t]\right\|^2 = \sum_{k=1}^K \boldsymbol{w}_k^2 \mathbb{E}\left\|\nabla\mathcal{L}_k - \mathbb{E}[\nabla\mathcal{L}_k]\right\|^2 + \sum_{k\neq j} \boldsymbol{w}_k \boldsymbol{w}_j \mathbb{E}\left[\langle\nabla\mathcal{L}_k - \mathbb{E}[\nabla\mathcal{L}_k], \nabla\mathcal{L}_j - \mathbb{E}[\nabla\mathcal{L}_j]\rangle\right].$$

**Proxy-Driven Weight Adjustment.** The proxy model optimizes weights via $\boldsymbol{w}_k^{t+1} \propto \boldsymbol{w}_k^t e^{\eta_k \ell_{\mathrm{dif}}^k}$, where $\ell_{\mathrm{dif}}^k = \mathcal{L}_k(\theta_{\mathrm{proxy}}) - \mathcal{L}_k(\theta_{\mathrm{ref}})$. Assuming local gradient variance $\mathbb{E}\|\nabla\mathcal{L}_k - \mathbb{E}[\nabla\mathcal{L}_k]\|^2 \leq \sigma_k^2$, the total variance satisfies:

$$\mathbb{E}\left\|g_{\mathrm{G}}^t - \mathbb{E}[g_{\mathrm{G}}^t]\right\|^2 \leq \sum_{k=1}^K \boldsymbol{w}_k^2 \sigma_k^2 + \sum_{k\neq j} \boldsymbol{w}_k \boldsymbol{w}_j \rho_{kj} \sigma_k \sigma_j,$$

where $\rho_{kj}$ is the gradient correlation coefficient between clients $k$ and $j$.

**Optimal Variance Reduction via Proxy Weights.** If the proxy adjusted weights satisfy $\boldsymbol{w}_k \propto \frac{1}{\sigma_k^2}$, we can construct the Lagrangian as:

$$\mathcal{L} = \sum_{k=1}^K \boldsymbol{w}_k^2 \sigma_k^2 - \lambda\left(\sum_{k=1}^K \boldsymbol{w}_k - 1\right),$$

take derivatives w.r.t. $\boldsymbol{w}_k$, and solve for optimal weights:

$$\boldsymbol{w}_k^* = \frac{1/\sigma_k^2}{\sum_{j=1}^K 1/\sigma_j^2} \quad\Rightarrow\quad \tilde{\sigma}^2 = \sum_{k=1}^K (\boldsymbol{w}_k^*)^2 \sigma_k^2 = \frac{1}{\sum_{k=1}^K \frac{1}{\sigma_k^2}}.$$

Though $\boldsymbol{w}_k \propto 1/\sigma_k^2$ is idealized, the proxy model approximates client reliability using $\ell_{\mathrm{dif}}^k$ (assuming $\sigma_k^2 \propto \ell_{\mathrm{dif}}^k$), leading to:

$$\tilde{\sigma}^2 \leq \max_k \frac{\sigma_k^2}{K} \cdot \frac{e^{2\eta_k \ell_{\mathrm{dif}}^k}}{\left(\sum_{k=1}^K e^{\eta_k \ell_{\mathrm{dif}}^k}\right)^2}.$$

This upper bound is strictly smaller than FedAvg's $\sigma_{\mathrm{FedAvg}}^2 = \frac{1}{K}\sum_{k=1}^K \sigma_k^2$ when $\eta$ is properly tuned.

**Step 3: Enhanced Convergence Bound.** By combining the refined descent lemma and the enhanced variance reduction facilitated by the proxy model, we derive a more robust bound on the convergence rate, effectively reducing errors and speeding up convergence compared to traditional methods.

$$\mathbb{E}[\mathcal{L}(\theta^T) - \mathcal{L}(\theta^*)] \leq \frac{\tilde{C}}{T} + \frac{\tilde{\sigma}^2}{P}\left(\frac{1}{K}\sum_{k=1}^K \frac{1}{n_k}\right) + \mathcal{O}(\epsilon).$$

This approach ensures a more adaptive and responsive FL environment, effectively addressing the complexities of real-world distributed learning scenarios.

## A.3 Theorem 3: Convergence of Proxy Model with Excess Loss

**Theorem A.3** (Convergence of Proxy Model). *Under the assumptions of L-smoothness and bounded gradient $\|\nabla \ell_{exc}^k\| \leq G$, the proxy model with task weights $\alpha_k$ updated by $\alpha_\tau^{t+1} \propto \alpha_\tau^t e^{\eta_\tau \ell_{exc}^\tau}$, satisfies:*

$$\mathbb{E}[\mathcal{L}(\theta^T) - \mathcal{L}(\theta^*)] \leq \frac{\tilde{C}_{proxy}}{T} + \frac{\tilde{\sigma}_{proxy}^2}{P}\left(\frac{1}{\mathcal{T}}\sum_{\tau=1}^{\mathcal{T}}(\alpha_\tau)^2\right) + \mathcal{O}(\epsilon).$$

*where $\tilde{C}_{proxy}$ is a constant reflecting the weight adjustments and $\tilde{\sigma}_{proxy}^2$ represents the reduced variance in the gradients influenced by the excess loss of proxy model and reference model, and $\mathcal{O}(\epsilon)$ captures the error due to the heterogeneity of the tasks among clients.*

**Proof of theorem 3.** The proxy model's convergence follows the same framework as the agent model (Theorem 2), with modified weights $\alpha_\tau^t$ that adapt to different tasks $\ell_{exc}^\tau$. The key adaptation lies in the variance term:

$$\text{Proxy Variance} = \frac{\tilde{\sigma}_{proxy}^2}{P}\left(\frac{1}{\mathcal{T}}\sum_{\tau=1}^{\mathcal{T}}(\alpha_\tau)^2\right) \leq \max_\tau \frac{\sigma_\tau^2}{\mathcal{T}} \cdot \frac{e^{2\eta_\tau \ell_{exc}}}{\left(\sum_{\tau=1}^{\mathcal{T}} e^{\eta_\tau \ell_{exc}^\tau}\right)^2}.$$

The full derivation parallels Theorem 2's steps, replacing client weights $w_k^t$ with task weights $\alpha_\tau^t$, and differential loss $\ell_{dif}$ with excess loss $\ell_{exc}$.

# B Supplementary Experiments

## B.1 Experimental Setup

The hierarchical architecture of FedRAM comprises three model categories: (1) *reference models* ($\theta_{ref}$) preserving task-specific knowledge through localized training, (2) *proxy models* ($\theta_{proxy}$) analyzing cross-task relationships via parameter ($\theta^*$), and (3) *agent models* ($\theta_{agent}$) integrating global knowledge through larger parameters ($\theta^+$). Our default training environment involves $K = 10$ clients handling $N = 7$ distinct tasks over $T = 50$ communication rounds. The optimization process employs a learning rate $\beta = 10^{-3}$ with weight decay $wd = 10^{-4}$ to prevent overfitting. Critical hyperparameters include an exponential scaling factor $\eta = 0.1$ for contribution reweighting and a smoothing parameter $s = 0.01$ to stabilize gradient updates. Notably, each communication round completes $P = 1$ local training step to minimize client-side computation overhead. This default experimental setting validates through experiment results in Section 5.2 and Appendix B.

Table 4: Default experimental settings. $\theta^*$ denotes the reference and proxy models, $\theta^+$ the agent model.

| Hyperparameters | Definition | Values |
|:---:|:---:|:---:|
| $\theta_{ref}$ | Reference Model | $\theta^*$ |
| $\theta_{proxy}$ | Proxy Model | $\theta^*$ |
| $\theta_{agent}$ | Agent Model | $\theta^+$ |
| $K$ | Number of Clients | 10 |
| $\mathcal{T}$ | Number of Tasks | 7 |
| $\beta$ | Learning Rate | 1E−3 |
| $wd$ | Weight Decay | 1E−4 |
| $T$ | Communication Rounds | 50 |
| $P$ | Training Steps | 1 |
| $\eta_{task}, \eta_{client}$ | Exponential Scaling Factor | 0.1 |
| $s$ | Smoothing Parameter | 0.01 |

## B.2 Data Initialization.

Figure 5 displays the heterogeneity in both task distribution and data availability across different federated clients, illustrating significant heterogeneity in the data handled by different clients. Panel

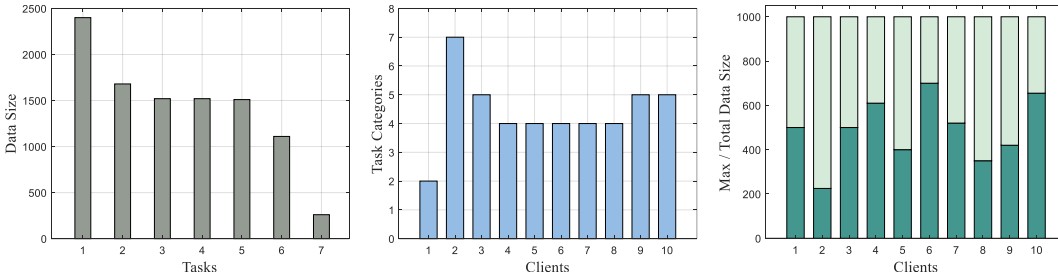

Figure 5: Heterogeneity Statistics Across Tasks and Clients. (a) Gray bars represent data size variations across different tasks. (b) Blue bars indicate the distribution of task categories per client, showing the diversity in tasks handled by different clients. (c) Green bars depict the data size within each client domain, where darker green represents the maximum dataset size, and lighter green indicates the total available data.

(a) highlights the imbalance in task sizes, where some tasks contain significantly more training samples than others, leading to disproportionate model updates during federated training. Panel (b) shows that clients engage with varying numbers of tasks, with some handling a diverse set of task categories while others specialize in fewer domains. This discrepancy suggests the necessity for flexible model adaptation strategies. Finally, panel (c) visualizes the data size per client, revealing that certain clients have access to large datasets but may not fully utilize them. The overall heterogeneity observed in task distribution and data volume underscores the limitations of traditional FL approaches, such as FedAvg, which assume homogeneous data distributions.

### B.3 Hyperparameter Analysis

**Effect of $\eta_{\text{client}}$ on Client Weights Evolution and Performance.** Figure 6 depicts how client weights evolve over iterations in FedRAM, reflecting adjustments based on the parameter $\eta_{\text{client}}$. The variation in $\eta_{\text{client}}$ values shows distinct convergence behaviours, which are critical for understanding how FedRAM adapts to different client needs. For values of $\eta_{\text{client}}$ near 0, weight changes are gradual and uniform, indicating a balanced approach to weight adjustments among clients. As $\eta_{\text{client}}$ moves away from 0, either negatively or positively, the weight trajectories show more pronounced disparities between clients, suggesting a more aggressive or conservative reweighting strategy. This is particularly noticeable with $\eta_{\text{client}}$ values of -1 and 1, where weights converge more distinctly.

Figure 8(b) further demonstrates how $\eta_{\text{client}}$ affects global performance, local adaptation, and out-of-domain generalization. Small values of $\eta_{\text{client}}$ result in a more stable and gradual improvement in accuracy, while extreme values lead to larger performance variations, reflecting different aggregation strategies under heterogeneous conditions. As $\eta_{\text{client}}$ moves further from 0 (either negative or positive), weight updates become more aggressive, amplifying disparities in client importance. This explains the increasing trend in global performance seen in Figure 6.

We analyze the impact of hyperparameter $\eta_{\text{client}}$ based on Figure 8(b). The results indicate that positive values of $\eta_{\text{client}}$ lead to higher weights being assigned to clients with larger losses, encouraging the global model to prioritize the underperforming models for improvement. In contrast, negative values of $\eta_{\text{client}}$ allocate greater weight to clients with smaller global losses, reinforcing the influence of well-performing models during global evaluation. Empirical observations in Figure 8(b) suggest that positive $\eta_{\text{client}}$ values yield more stable and consistent performance across all four evaluation metrics, making them a preferable choice for balancing model adaptation and generalization.

**Effect of $\eta_{\text{task}}$ on Task Weights Evolution and Performance.** Figure 7 illustrates the dynamic evolution of task weights over different training iterations in FedRAM, showing how the method adapts task weights based on their contribution to global model performance. As training progresses, the task weights shift, with lighter blue shades representing earlier training rounds and darker shades indicating later stages. The red line reflects the initial task weight distribution, which is influenced by the dataset size ratio (data size to total data size). The gradual adjustments highlight that FedRAM has the ability to iteratively prioritize tasks based on their relevance and performance, allowing the model to better handle task heterogeneity. This dynamic weight adjustment process, driven by the proxy and agent models, ensures that tasks with smaller datasets or less initial contribution are given higher

weights in later stages, improving the overall performance. The task-wise reweighting mechanism is a key strength of FedRAM.

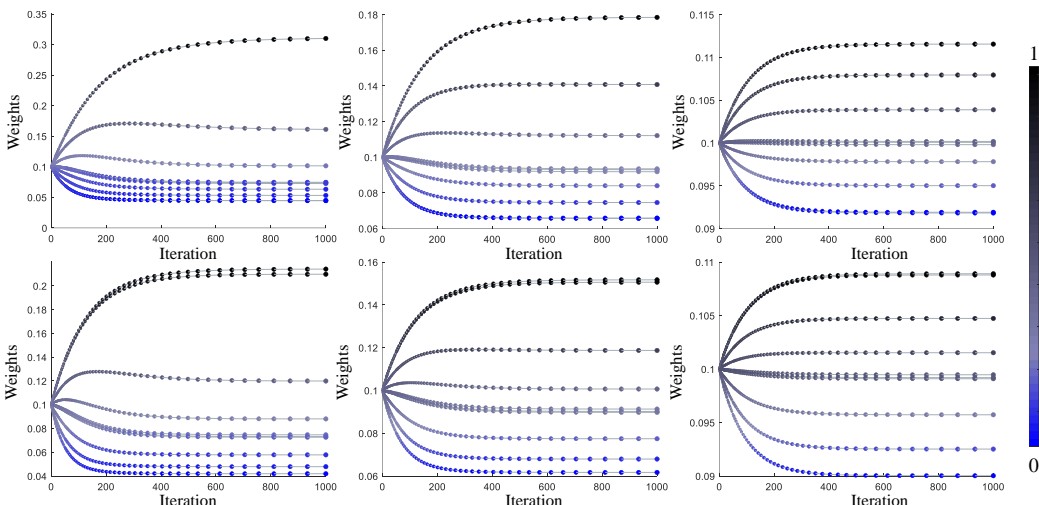

Figure 6: Evolution of Client Weights Over Training Iterations in FedRAM. The subfigures correspond to $\eta_{\text{client}} = -1, -0.5, -0.1, 0.1, 0.5, 1$, respectively. The parameter $\eta$client results in varying convergence behaviours while preserving the relative weight relationships among clients. Darker points indicate higher weight values, and blue points represent lower values.

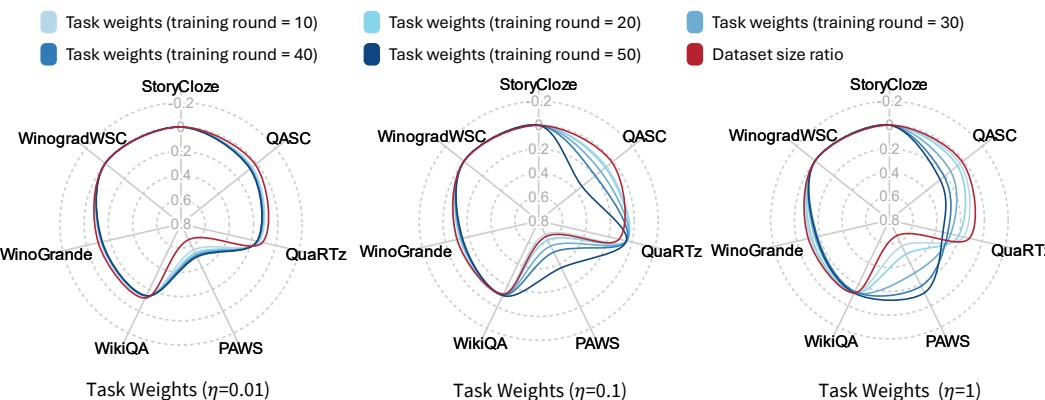

Figure 7: Evolution of Task Weights Over Training Iterations in FedRAM. The radar charts illustrate the dynamic adjustment of task weights under different $\eta_{\text{task}}$ settings. The red line represents the initial distribution of task weights (data size / total data size). The lines in blue tones present the evolution of task weights as training progresses from early (lighter shades) to later (darker shades) stages.

## B.4 Model Scalability

**Effect of Agent-to-Proxy (A/P) Ratio on Model Performance.** Figure 8(a) presents the impact of the Agent-to-Proxy (A/P) ratio, a concept introduced in this study to evaluate the feasibility of using a smaller proxy model to optimize a larger agent model. The A/P ratio is defined as:

$$\text{A/P Ratio} = \frac{\text{Number of Agent Parameters}}{\text{Number of Proxy Parameters}} \tag{15}$$

A lower A/P ratio suggests a relatively larger proxy model, which may improve stability but increase computational overhead. Conversely, higher A/P ratios imply a larger agent model trained with fewer proxy parameters, allowing more efficient adaptation while maintaining competitive performance.

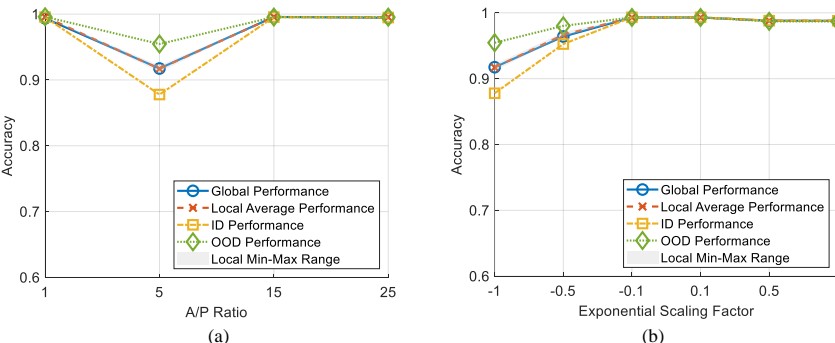

Figure 8: Impact of Hyperparameters on FedRAM Performance. (a) Effect of the agent-to-proxy (A/P) ratio, which controls the relative size of the agent and proxy models, on performance. (b) Effect of the exponential scaling factor, which influences weight adjustments during aggregation, on performance stability.

Table 1 in the main text and Table 6 provide empirical results for different A/P ratios. Table 1 corresponds to an A/P ratio of 15, while Table 6 corresponds to an A/P ratio of 25. Comparing these two settings, we observe that a higher A/P ratio leads to an improvement in global performance (82.16 vs. 79.74) and local adaptation (83.22 vs. 79.62), suggesting that FedRAM benefits from a well-tuned proxy-to-agent parameter balance. However, a larger A/P ratio can also lead to slightly reduced stability in out-of-domain generalization, as seen in the marginal drop in OOD performance (82.89 vs. 82.67).

These results demonstrate that using a smaller proxy model to guide the reweighting of a larger agent model is entirely feasible. While increasing the A/P ratio improves performance, the effectiveness of FedRAM remains strong even when the proxy model is significantly smaller, highlighting the efficiency of our proposed approach in leveraging lightweight models for scalable optimization.

**Computational Cost Analysis.** As shown in Table 5, the reference model, proxy model, and agent model exhibit significantly differentiated computational resources consumption. The agent model $\theta_{\text{agent}}$ persistently dominates system resources, consuming $85.08\% - 88.02\%$ of total computation time across client scales ($N =10$ to $50$). Notably, the stability of component-wise proportions underscores FedRAM's scale-invariant resource allocation design. These findings quantitatively validate FedRAM's architectural robustness for federated learning systems requiring adaptive resource-aware coordination.

Table 5: Computational cost of FedRAM components.

| Algorithm | Model | $N = 10$ | | $N = 20$ | | $N = 50$ | |
|---|---|---|---|---|---|---|---|
| | | Time (s) | Proportion (%) | Time (s) | Proportion (%) | Time (s) | Proportion (%) |
| FedRAM | $\theta^*_{\text{ref}}$ | 120 | 1.00 | 720 | 1.45 | 1093 | 1.10 |
| | $\theta^*_{\text{proxy}}$ | 1680 | 13.92 | 5231 | 10.49 | 10750 | 10.88 |
| | $\theta^+_{\text{agent}}$ | 10260 | 85.08 | 43903 | 88.06 | 87000 | 88.02 |
| | $\sum \theta$ | **12060** | 100.00 | **49854** | 100.00 | **98843** | 100.00 |

**Performance of Larger Agent Models.** We validate that the proposed FedRAM method significantly outperforms existing FL-MTL approaches in various evaluation metrics. Table 6 presents a comparative analysis of FedRAM against several SOTA FL methods across diverse NLP tasks. We highlight the following key observations:

(i) Superior Global Average and Task-wise Performance: FedRAM achieves a global F1-score of 82.16, outperforming all baseline methods. In particular, FedRAM attains the highest F1-score in two critical tasks, Story Cloze (90.77) and wsc (78.91), demonstrating its efficacy in handling various NLP challenges. Compared to FedMTL, which excels in specific tasks but struggles with broader generalization, FedRAM achieves a more balanced and robust performance across tasks.

Table 6: F1-Score Comparison of FL Methods Performed on Larger Models. This table shows the F1-scores for different FL methods across multiple NLP tasks, including global F1-scores and local F1-scores within the bottom decile, alongside ID and OOD evaluations. Scores in **Bold** indicate the best performance, while scores underlined denote the second best.

| Methods | Tasks | | | | | | | Global F1-Score | Local F1-Score / Bottom Decile | ID / OOD Evaluation |
| --- | --- | --- | --- | --- | --- | --- | --- | --- | --- | --- |
| | PAWS | WSC | Wino Grande | QASC | QuaRTz | Story Cloze | WikiQA | | | |
| FedAvg [1] | 88.53 | 70.90 | 79.88 | 32.50 | 80.14 | 80.87 | 79.79 | 80.75 | 81.33 / 79.58 | 81.18 / 81.51 |
| FedProx [10] | 77.27 | 74.90 | 78.70 | **38.26** | 80.37 | 78.65 | 65.73 | 74.79 | 78.55 / 74.89 | 78.06 / 78.97 |
| Ditto [11] | 86.06 | 72.38 | 77.28 | 31.60 | **80.75** | 80.24 | 79.79 | 79.68 | 80.70 / 79.08 | 80.02 / 81.59 |
| FedRep [27] | 88.79 | 72.11 | **80.71** | 32.78 | 79.12 | 82.79 | 63.87 | 80.72 | 81.25 / 79.06 | 81.59 / 78.91 |
| DBE [8] | 88.52 | 70.47 | 79.01 | 31.88 | 80.09 | 81.58 | 79.79 | 80.66 | 81.12 / 77.76 | 81.14 / 82.06 |
| FedMTL [28] | **90.26** | 73.67 | 79.21 | 33.06 | 69.72 | 88.85 | 79.79 | 78.00 | 82.33 / 77.55 | 81.92 / 82.67 |
| FedBone [15] | 75.26 | 74.78 | 73.93 | 30.83 | 76.98 | 73.90 | 79.78 | 76.61 | 78.23 / 75.60 | 76.24 / 82.51 |
| FedRAM (Ours) | 90.00 | **78.91** | 78.36 | 33.03 | 80.70 | **90.77** | 79.79 | **82.16** | **83.22 / 80.63** | **83.35 / 82.89** |

(ii) Enhanced Performance in Low-performing Clients: In local F1-score evaluation within the bottom decile, FedRAM reaches 83.22, surpassing the second-best method (FedMTL, 82.33) by 0.89. This highlights FedRAM's ability to mitigate performance degradation in lower-performing clients. The reweighting mechanism introduced in FedRAM effectively balances task importance, ensuring robust personalization across heterogeneous data distributions.

(iii) Effective Handling of Statistical Heterogeneity: Compared to FedAvg and FedProx, which do not incorporate explicit mechanisms to address inter-task conflicts, FedRAM improves performance across multiple tasks by dynamically adjusting task sample rates and client aggregation weights. For instance, on the WSC task, FedRAM achieves 78.91, surpassing the second-best score (FedProx, 74.90) by a significant margin, demonstrating its capability to address heterogeneity.

(iv) In-Domain Personalization and Out-of-Domain Generalization: FedRAM demonstrates superior performance in both in-domain personalization and out-of-domain generalization, outperforming baseline methods across various NLP tasks. For ID evaluations, FedRAM achieves the highest score (83.35), showing its ability to personalize well within the federated learning setting. Meanwhile, for OOD generalization, it also achieves the highest score (82.89), demonstrating its ability to adapt to unseen data distributions better than other methods. Specifically, while FedMTL exhibits competitive OOD performance (81.92), its overall global F1-score (78.00) is lower, indicating a trade-off between generalization and global performance. Similarly, FedAvg, which assumes homogeneous data distributions, underperforms in ID and OOD settings, reinforcing the importance of adaptive reweighting strategies.

(v) Performance Improvement with Larger Models: Comparing Tables 1 and 6, we observe that using a larger model improves overall performance across all methods. Notably, FedRAM's global F1-score increases from 75.94 to 82.16, and its local F1-score (bottom decile) improves from 79.62 to 83.22. Additionally, its ID / OOD evaluation scores rise from 72.89 / 76.32 to 83.35 / 82.89. These enhancements suggest that FedRAM benefits significantly from increased model capacity, further reinforcing its effectiveness in handling complex multi-task federated learning scenarios.

Overall, FedRAM establishes itself as a robust and efficient FL-MTL framework by effectively mitigating inter-task conflicts and optimizing both personalization and generalization. Compared to existing personalization strategies, including DBE and FedBone, FedRAM achieves at least a 3% improvement across multiple metrics, validating the effectiveness of its novel reweighting mechanism. This makes FedRAM a well-rounded approach for federated learning scenarios with heterogeneous and multi-task data.

## B.5 Client Scalability

As the number of participating clients increases, global performance (blue) shows a slight downward trend, likely due to increased data heterogeneity. However, local average performance (orange) and ID performance (yellow) remain relatively stable, indicating that FedRAM effectively adapts to the additional heterogeneity introduced by more clients. Particularly, OOD performance (green) remains consistently high, demonstrating strong generalization capabilities. The local min-max range (green

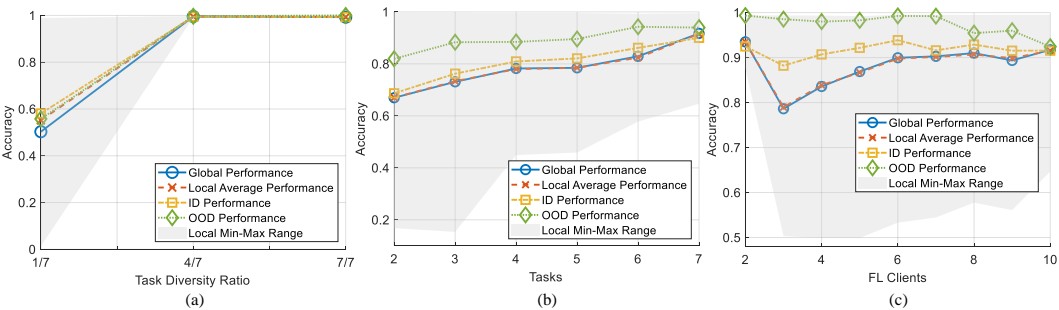

Figure 9: Impact of Heterogeneity on FedRAM Performance. (a) Performance trends across varying task diversity ratios. (b) Accuracy results across different numbers of tasks. (c) Model performance with an increasing number of FL clients.

diamonds) widens slightly, suggesting some variability in individual client performance, but overall stability is maintained.

Table 7: Comparison of FL Methods with client $K = 20$ and $50$. This table shows the F1-scores for different FL methods across multiple NLP tasks, including global F1-scores and local F1-scores, alongside ID and OOD evaluations. Scores in **Bold** indicate the best performance.

| Clients | Methods | Tasks | | | | | | | Global / Local F1-Score | ID / OOD Evaluation |
| | | PAWS | WSC | Wino Grande | QASC | QuaRTz | Story Cloze | WikiQA | | |
| --- | --- | --- | --- | --- | --- | --- | --- | --- | --- | --- |
| $K = 20$ | FedAvg [1] | 75.09 | 72.69 | 61.35 | 34.16 | 79.10 | 72.20 | 71.13 | 67.98 / 68.67 | 70.07 / 67.31 |
| | Ditto [11] | 75.09 | 73.63 | 61.05 | 30.88 | 78.79 | 80.84 | 71.13 | 70.25 / 72.98 | 75.85 / 72.65 |
| | FedMTL [28] | 74.48 | 72.16 | 72.05 | 32.10 | 73.89 | **83.14** | 71.13 | 69.10 / 70.35 | 75.78 / 73.33 |
| | FedBone [15] | 69.54 | 66.90 | 70.11 | 33.05 | 58.88 | 76.22 | 71.13 | 66.91 / 66.85 | 69.43 / 68.55 |
| | FedRAM (Ours) | **75.69** | **73.63** | **74.11** | **36.93** | 79.54 | 78.22 | 71.13 | **70.86 / 73.33** | **76.31 / 74.54** |
| $K = 50$ | FedAvg [1] | 72.75 | 69.83 | 65.44 | 33.46 | 69.52 | 71.50 | 69.79 | 64.79 / 66.33 | 67.10 / 65.53 |
| | Ditto [11] | 72.75 | 69.83 | 65.44 | 33.46 | 69.52 | 71.50 | 69.79 | 65.91 / 67.83 | 68.50 / 67.24 |
| | FedMTL [28] | 68.11 | 73.67 | 63.08 | 32.11 | 67.56 | 70.90 | 69.79 | 63.28 / 65.37 | 63.86 / 62.13 |
| | FedBone [15] | 69.78 | 70.41 | 68.18 | 34.33 | 63.51 | 73.78 | 69.79 | 64.53 / 66.29 | 65.08 / 65.77 |
| | FedRAM (Ours) | **73.88** | **74.27** | **68.76** | **35.78** | **73.90** | **75.53** | 69.79 | **67.52 / 68.85** | **70.89 / 68.90** |

## B.6 Task Scalability

Building on the observations from Figure 5, Figure 9 presents the impact of task and data heterogeneity on the performance of FedRAM.

(a) **Task Diversity Ratio.** As the task diversity increases, global performance (blue) and local average performance (orange) both improve significantly, indicating that FedRAM benefits from more diverse task distributions. However, the ID performance (yellow) and OOD performance (green) maintain a relatively stable trend, suggesting that task diversity does not compromise the model's generalisation ability. The local min-max range (green diamonds) shows relatively low variance, indicating consistent performance across different clients.

(b) **Number of Tasks.** As the number of tasks increases, global performance (blue) and local average performance (orange) gradually improve, but the improvement becomes marginal beyond five tasks. ID performance (yellow) follows a similar increasing trend, while OOD performance (green) remains consistently high. The local min-max range shows a steady spread, implying that FedRAM maintains robust client-wise performance even with more tasks.

(c) **Extreme Heterogeneity.** In Table 8, we present the results in *Setting 2* and assign one client with one distinct task. The improvements in F1-scores derive from the cut down in client number $K = \mathcal{T} = 7$ in this setting. The results validate FedRAM's adaptive ability to extreme heterogeneous conditions.

(d) **Computer Vision Tasks.** We present a computer vision (CV) task compatibility study with FedRAM in Table 9. We base our experiments on seven distinct CV datasets: MNIST, EuroSat,

Table 8: F1-Score Comparison of FL Methods Performed on Larger Models. This table shows the F1-scores for different FL methods across multiple NLP tasks, including global F1-scores and local F1-scores within the bottom decile, alongside ID and OOD evaluations. Scores in **Bold** indicate the best performance, while scores underlined denote the second best.

| Methods | Tasks | | | | | | | Global / Local F1-Score | ID / OOD Evaluation |
|---|---|---|---|---|---|---|---|---|---|
| | PAWS | WSC | Wino Grande | QASC | QuaRTz | Story Cloze | WikiQA | | |
| FedAvg [1] | 88.45 | 90.32 | 84.33 | 76.11 | 79.53 | 84.30 | 90.29 | 82.67 / 87.96 | 90.29 / 84.78 |
| Ditto [11] | 89.51 | 88.86 | **90.05** | 76.13 | 83.42 | 83.44 | 89.70 | 83.85 / 88.67 | 90.45 / 87.03 |
| FedMTL [28] | **90.48** | 82.37 | 80.07 | **82.68** | 89.54 | 83.14 | 81.88 | 83.88 / 88.48 | 90.53 / 86.51 |
| FedBone [15] | 89.25 | 83.47 | 84.79 | 75.90 | 88.74 | 88.64 | 85.50 | 84.06 / 89.53 | 91.02 / 86.95 |
| FedRAM (Ours) | 90.36 | **90.32** | 87.94 | 78.48 | **90.40** | **90.50** | **90.42** | **84.29 / 90.29** | **91.53 / 87.26** |

Fashion-MNIST, SVHN, DTD, Stanford-Cars, and CIFAR-10. We set reference models and proxy models as Swin Transformer, while the agent models are the CLIP-ViT. Other settings followed the condition in Appendix B.1.

Table 9: Comparison of FL Methods Performed on Vision Tasks. Scores in **Bold** indicate the best performance.

| Methods | Tasks | | | | | | | Global / Local F1-Score | ID / OOD Evaluation |
|---|---|---|---|---|---|---|---|---|---|
| | MNIST | EuroSat | Fashion -MNIST | SVHN | DTD | Stanford -Cars | CIFAR-10 | | |
| FedAvg [1] | **97.22** | 38.13 | 81.88 | 66.14 | 20.08 | 80.62 | 77.47 | 78.09 / 76.36 | 68.11 / 62.57 |
| FedProx [11] | 47.35 | 8.33 | 27.30 | 58.61 | 9.89 | 57.92 | 54.20 | 50.73 / 44.67 | 36.67 / 28.41 |
| DBE [28] | 97.22 | 24.10 | 76.67 | 65.87 | 20.21 | **80.79** | 76.48 | 76.93 / 75.66 | 64.71 / 58.33 |
| OBF [30] | 96.07 | 4.03 | **87.79** | **78.79** | 33.14 | 74.77 | 79.35 | 78.84 / 65.18 | 53.34 / 47.96 |
| FedGF [29] | 72.73 | 5.19 | 75.03 | 30.08 | 27.40 | 71.72 | 75.79 | 75.08 / 62.88 | 52.54 / 44.04 |
| FedRAM (Ours) | **97.22** | **50.55** | 87.70 | 78.49 | **43.28** | 80.28 | **80.91** | **82.05 / 79.23** | **70.94 / 65.26** |

