# OpenReview forum: "FedRAM: Federated Reweighting and Aggregation for Multi-Task Learning"
_NeurIPS.cc/2025/Conference — NeurIPS 2025 poster_

### Official Review · Reviewer_7drK · 2025-07-02

**Clarity:** 4
**Significance:** 3
**Originality:** 3
**Rating:** 5
**Confidence:** 4

**Summary:**

The paper presents FedRAM, a three-stage framework for heterogeneous federated multi-task learning.

Step 1 – Local Reference Model ($\theta^{ref}$): Each client trains a lightweight model on its own mixed-task data to capture the local distributions.

Step 2 – Proxy model ($\theta^{proxy}$): Clients and the server jointly train another lightweight global model with task-specific weights $\alpha$ that are updated from an excess-loss signal relative to the frozen reference. The server also derives client aggregation weights $w$ from the differential loss between local and global proxy performance.

Step 3 – Agent model ($\theta^{agent}$): a larger shared model is fine-tuned with data resampled according to $\alpha$ and aggregated according to $w$.

Experiments on seven NLP tasks with 10 clients demonstrate that FedRAM improves global F1, reduces the number of convergence rounds, and reduces total training time while maintaining both in-domain and out-of-domain accuracy gains.

**Questions:**

1. **Scalability to larger/backbone models:** Can FedRAM handle $\ge$ 1 B-parameter transformers or vision encoders? A single ImageNet-100 or GLUE-MNLI run would support significance.

2. **Robustness to partial participation:** How do $\alpha$ and $w$ adapt when only a random subset of clients join each round?

3. **Hyperparameter guidelines:** Figures 6 & 7 show strong sensitivity to $\eta_{client}$ and $\eta_{task}$. Could you propose a heuristic (e.g., set $\eta \propto 1/\sqrt{rounds}$) that works across datasets?

Criteria for score change: convincing large-backbone results and answering all questions would raise the score to 5. Evidence of instability under partial participation would decrease to 3.

**Ethical Concerns:**

["NO or VERY MINOR ethics concerns only"]

**Final Justification:**

The authors provide thorough and convincing answers and new evidence that addresses all my concerns.

I consider revising the score as these evidence meaningfully improves the paper’s significance and supports acceptance.

**Limitations:**

The authors acknowledge compute requirements and manual tuning of the energy parameters, but do not address: (i) privacy/fairness risks of task reweighting, (ii) runtime/energy overhead per stage, and (iii) lack of evidence on large-scale or non-NLP tasks. Clarifying these would strengthen the limitations section.

**Paper Formatting Concerns:**

There is no concern.

**Quality:**

3

**Strengths And Weaknesses:**

**Strengths**

Well-structured paper, intuitive Figures, and a running example showing the three roles. Detailed appendix with proofs and extra ablations.

Clear algorithm description with pseudocode and convergence proofs for the weight-adapted FedRAM and its proxy stage.

Comprehensive evaluation on six baselines, multiple metrics (global/local, ID/OOD), convergence curves, ablations, and hyperparameter studies.

This paper addresses a significant gap: the simultaneous task-level and client-level reweighting in FL-MTL, which leads to faster convergence and fairer performance. The hierarchical reference–proxy–agent idea could inspire other resource-aware FL designs.

Novel combination of proxy-mediated task weights and differential-loss-based client weights within a unified three-stage pipeline; distinct from prior task-correlation or model-merging works.

**Weaknesses**

All experiments use T5-small/base and only NLP tasks; vision or larger-scale models have not been tested. No robustness tests against adversarial or partially participating clients.

Improvements over the FedMTL are modest on some tasks; gains in ID/OOD may diminish for very heterogeneous or large-task suites. Practical adoption needs validation on larger models and non-text modalities.

---

> ### Author Rebuttal · Authors · 2025-07-31
>
> We thank the reviewer for the feedback. 7drK notes that “Novel combination of proxy-mediated task weights and differential-loss-based client weights within a unified three-stage pipeline” and “well-structured”, with “Comprehensive results”. We address specific questions below:
>
> ---
>
> ### **Q1: Scalability to larger/backbone models:**
> **Can FedRAM handle $\geq$ 1 B-parameter transformers or vision encoders? A single ImageNet-100 or GLUE-MNLI run would support significance.**
>
> **A1:** Thank you for this important question regarding the scalability of FedRAM to larger models and standard benchmarks.
>
> **Large Language Model Experiments**: We have conducted experiments with Llama-3.2-1B to demonstrate FedRAM's scalability to billion-parameter models. The setup aligns with **Section 5.1**.
>
> | Methods | $\theta_\text{ref}$ and $\theta_\text{proxy}$|$\theta_\text{agent}$| Global | Local | ID   | OOD  |
> |---------|---------|--------|------|------|------|------|
> | FedAvg |$\textit{None}$| LLama 1.24B| 84.20   | 86.08  | 86.32 | 83.69 |
> | FedRAM |bert 110M| LLama 1.24B| 84.43   | 89.64  | 91.78 | 87.40 |
> | FedRAM |T5-large 770M | LLama 1.24B | 86.77   | 92.03  | 92.24 | 88.19 |
>
> Due to computational resource and time constraints, we focused on existing datasets and comparison to Fedavg. From the table, FedRam can handle $\geq$ 1B models with consistent results across the four metrics (still **outperforms** FedAvg).
>
> **Vision Encoder Experiments**: For vision encoders, please refer to Appendix B.6, where we present experiments on several independent datasets using large-scale vision models. Specifically, we employed Swin Transformer as both $\theta_{\text{ref}}$ and $\theta_{\text{proxy}}$ and CLIP-ViT as $\theta_{\text{agent}}$.
>
>
> ### **Q2: Robustness to partial participation:**
> **How do $\boldsymbol{\alpha}$ and $\boldsymbol{w}$ adapt when only a random subset of clients join each round?**
>
> **A2:**
>
> Setup: We base the experiments on the same task dataset, clients and model setting in the main text **Section 5.1**.
> Within one communication round, we set participation ratio of clients to be 70% and 50%, which equals to 30% and 50% dropout rate, respectively. Under different dropout rate, we randomly select the clients to drop. Note that the task weights and client weights of the dropout clients are frozen for no FL training performed.
>
> We set the dropout rate to be 50% for Exp 1 and Exp 2. From Exp 1, it is observed that $\boldsymbol{\alpha}$ is robust. For instance, the task weights in Client 2 are stable from round 30 to 50.
>
> Exp 1: $\boldsymbol{\alpha}$ adaptation, here we take client 2's adaptation as an example:
> | Task  | round 0  | round 10 | round 20 | round 30   | round 40  | round 50  |
> |---------|---------|--------|-------|------|------|-|
> | 1 | 0.155  |  0.155  | 0.155  | 0.155 | 0.154  | 0.154  |
> | 2 | 0.225  |  0.210  | 0.209  | 0.207 | 0.205 | 0.205 |
> | 3 | 0.120 | 0.111   |  0.111 |  0.111 | 0.112 | 0.111 |
> | 4 | 0.120 |  0.123  |  0.125 | 0.125 | 0.126 | 0.126 |
> | 5 | 0.120 |  0.123  | 0.124  | 0.124 | 0.124 |  0.124 |
> | 6 | 0.140 | 0.154   | 0.153  | 0.155 | 0.154 | 0.155 |
> | 7 | 0.120 | 0.124   | 0.123  | 0.123 | 0.124 | 0.124 |
>
> In Exp 2, we present the client weights $\boldsymbol{w}$ across communication rounds. It is observed that $\boldsymbol{w}$ changes slowly at the beginning for suffering from client heterogeneity. The client weights begin to converge from round 30 and exhibit robustness.
>
> Exp 2: $\boldsymbol{w}$ adaptation:
> | Client  | round 0  | round 10 | round 20 | round 30   | round 40  | round 50  |
> |---------|---------|--------|-------|------|------|-|
> | 1 | 0.100  | 0.105 | 0.142  | 0.160 | 0.166  | 0.167  |
> | 2 | 0.100  | 0.099 | 0.092  | 0.089 | 0.088 | 0.087 |
> | 3 | 0.100 |  0.099   |  0.090 |  0.088 | 0.086 | 0.086 |
> | 4 | 0.100 |  0.099  |  0.095 | 0.093 | 0.092 | 0.092 |
> | 5 | 0.100 |  0.100  | 0.101  | 0.102 |  0.099 | 0.099 |
> | 6 | 0.100 | 0.100   | 0.097  | 0.095 | 0.094 |  0.094 |
> | 7 | 0.100 | 0.100   | 0.097  | 0.095 | 0.095 |0.095 |
> | 8 | 0.100 |  0.099  | 0.094  | 0.091 | 0.090 | 0.090 |
> | 9 | 0.100 | 0.099   | 0.092  | 0.089 | 0.090 | 0.090 |
> | 10 | 0.100 | 0.100 | 0.100  | 0.098 | 0.100 | 0.100 |
>
> Exp 3: We compare the partial clients participation in FedRAM and FedAvg:
> | Methods  | Global | Local | ID   | OOD  | Convergence Round |
> |---------|---------|--------|-------|------|------|
> | FedRAM|  75.94 | 79.62 | 72.89 | 76.32 | 18 |
> | FedRAM (Dropout Rate = 0.3) |  75.03 | 79.65 | 72.97 | 76.93 | 20 |
> | FedRAM (Dropout Rate =0.5)  | 73.67  | 76.88 | 70.92 | 74.71 | 24 |
> | FedAvg|  72.71 | 76.34 | 71.68 | 75.66 | 21 |
> | FedAvg (Dropout Rate = 0.3) |  72.54 | 74.33 | 70.67 | 75.68 | 24 |
> | FedAvg (Dropout Rate =0.5)  | 70.39  | 73.40 | 66.58 | 70.33 | 30 |
>
> From Exp 3, FedRAM outperforms the FedAvg baseline under the same dropout rate, demonstrating the robustness of FedRAM. Especially, it is noticed that FedRAM with even 0.5 dropout rate outperforms FedAvg with no dropout rate (full participation).  The main reason is that FedRAM performs better when less clients are participated. To confirm this, we refer to **Appendix B5** Figure 9(c). FedRAM shows an increase as the FL client number goes down, demonstrating FedRAM performs better when focusing on a smaller group of clients and performs more fine-grained reweighting.
> Besides, from Exp 2, we can observe that FedRAM upweights client 1 by 67%, which is effectively reflected on the metrics.
>
>
> ### **Q3: Hyperparameter guidelines:**
> **Figures 6 & 7 show strong sensitivity to $\eta_\text{client}$ and $\eta_\text{task}$. Could you propose a heuristic (e.g., set $\eta \propto 1/\sqrt{rounds}$ ) that works across datasets?**
>
> **A3:**
> 1. To determine the exponential scaling factors, we employed a logarithmic grid search strategy, ranging from 1E-2 to 1E2.  Our systematic analysis identified the optimal logarithmic value as 1E-1, with the effective parameter range being $[-1, 1]$, as demonstrated in Figure 8. Within this range, the impact on global framework performance remains consistently stable.
> 2. Take $\eta_\text{task}$ as an example. Theoretically, assuming $\theta_\text{ref}$ in Step 1 is well trained, and the loss $\ell^\tau_\text{ref}$ equals to 0. The term $\ell^\tau_\text{exc}$ equals to $\frac{\ell_{\text{proxy}}^\tau}{{|\mathcal{D}^\tau_L|}}$. Assuming $\ell^\tau_\text{proxy}$ goes down per iteration. We make sure the first-iteration of $\alpha^1_\tau\leftarrow\alpha^0_\tau e^{ \eta_\text{task} \ell^\tau_\text{exc} } \leq 1$ should not excess the natural upper bound 1 (note that $\alpha^0_\tau=\frac{1}{K}$). Therefore, it is less risky to set  $\eta_\text{task}\leq\frac{\text{ln}K}{\ell^\tau_\text{exc} }=\frac{|\mathcal{D}^\tau_L| \times\text{ln}K}{\ell^\tau_\text{proxy}}$, where $|\mathcal{D}^\tau_L|$ is denoted as the training dataset size., $K$($K>1$) as the number of clients, $\ell^\tau_\text{proxy}$ as the maximum value of first step proxy loss performed on task $\tau$.
> 3. We do not recommend extensive hyperparameter tuning for these parameters. For practical implementations, we recommend using $\eta = 1, 0.1$ or $0.01$ as simple and effective default values.

---

> > ### Comment · Reviewer_7drK · 2025-08-04
> >
> > Thank you so much for providing thorough answers to my concerns and questions.
> >
> > Additional experiments provided for scalability and partial client participation are convincing and addressed my points.
> >
> > The practical hyperparameter guidelines supported by theoretical analysis, addressing concerns about sensitivity and tuning.
> >
> > I consider revising the score as the new evidence meaningfully improves the paper’s significance and supports acceptance.

---

> ### Author Response · Authors · 2025-08-04
> **Thank you**
>
> Dear Reviewer 7drK,
>
> Thank you very much for your thoughtful and constructive feedback, and for taking the time to reconsider our paper after reviewing our responses.
>
> Best regards,
>
> Authors of Paper 6648

---

### Official Review · Reviewer_9ir2 · 2025-07-02

**Clarity:** 3
**Significance:** 3
**Originality:** 3
**Rating:** 4
**Confidence:** 3

**Summary:**

This paper introduces FedRAM, a federated multi-task learning (FL-MTL) framework that addresses statistical heterogeneity, task interference, and imbalanced client contributions. FedRAM proposes a three-stage design involving:

Local reference models to capture client-specific distributions,

Lightweight proxy models to dynamically tune task and client weights based on excess and differential losses,

Agent models to aggregate using learned weights.

The authors demonstrate strong performance across six NLP datasets under both in-domain (ID) and out-of-domain (OOD) settings, with improvements in convergence speed and communication efficiency. The paper includes convergence proofs and extensive experiments.

**Questions:**

See above

**Ethical Concerns:**

["NO or VERY MINOR ethics concerns only"]

**Final Justification:**

Thank for authors' rebuttal, which addressed my concerns. I will maintain my positive attitude and score.

**Limitations:**

Yes, limitations are reasonably addressed in Section 5.2.5 and Appendix B. However, the main paper should bring some of these forward, especially:

Sensitivity to hyperparameters,

Dependency on accurate proxy training,

Lack of support for non-transformer tasks.

**Paper Formatting Concerns:**

Conforms to NeurIPS formatting. Figures and tables are legible.

**Quality:**

3

**Strengths And Weaknesses:**

Strengths:

The paper offers a well-structured and modular solution (reference, proxy, agent) that aligns with real-world FL-MTL settings.

Experiments are comprehensive, covering ablation studies, scalability, convergence speed, computational costs, and performance on both ID and OOD tasks.

The method is evaluated against a diverse set of recent and relevant baselines, such as FedMTL, FedRep, FedBone, and DBE.

Weaknesses:

The exponential update rules for task and client weights (Equations 7, 11) are intuitive but not enough motivated. Why exponential over other update rules?

The authors decouple the optimization into three phases, but interactions among these models are not sufficiently validated. What happens when proxy and reference disagree? Is the proxy model actually learning meaningful generalization signals?

All experiments are based on T5-small/base models. While practical, it leaves questions on generality across modalities (e.g., vision, speech) or architectures (e.g., CNNs, GNNs).

The choice of η_task, η_client, smoothing factor s and their effects are not discussed in the main text. These seem critical to convergence and stability.

While the related work section is informative, it would benefit from a more comprehensive coverage of relevant baselines, especially in the areas of personalized federated learning and personalized device-cloud collaboration. The experimental section could also be enhanced by including these baselines, if time allows.

---

> ### Author Rebuttal · Authors · 2025-07-31
>
> We thank the reviewer for the feedback. 9ir2 notes that “Novel combination of proxy-mediated task weights and differential-loss-based client weights within a unified three-stage pipeline” and “well-structured”, with “Comprehensive results”. We address specific questions below:
>
> ---
>
> ### **W1: The exponential update rules for task and client weights (Equations 7, 11) are intuitive but not enough motivated. Why exponential over other update rules?**
>
> **A1:** The exponential update rules follow established principles in distributionally robust optimization, particularly the multiplicative weights framework used in group DRO literature [1]. Exponential updates provide convergence guarantees and numerical stability when dynamically balancing competing objectives across different groups or tasks, especially in the non-iid fl setting.
>
> Unlike additive updates, exponential scaling prevents poorly-performing tasks/clients from being completely ignored while avoiding dominance by any single component. The exponential form ensures that weights remain positive (essential for valid probability distributions) while providing responsive adaptation to performance changes.
>
> [1] Sagawa et al. Distributionally robust neural networks for group shifts. ICLR 2020.
>
> ### **W2: The authors decouple the optimization into three phases, but interactions among these models are not sufficiently validated. What happens when proxy and reference disagree? Is the proxy model actually learning meaningful generalization signals?**
>
> **A2:** We appreciate the reviewer's concern about model interactions. To clarify, the proxy model in FedRAM is not designed to learn generalization signals but serves as a hyperparameter learning mechanism for the subsequent agent model training.
>
> **Proxy Model's Specific Role (Figure 3):**
> As illustrated in Figure 3, the proxy model's sole purpose is to produce two sets of hyperparameters: task weights ($\alpha_\tau$) and client aggregation coefficients ($w_k$). After Step 2 completes, the proxy model itself is discarded, and only these learned weights are passed to Step 3 for resampling and aggregation in agent model training.
>
> **Handling Reference-Proxy Disagreements:**
> This disagreement is constructively managed through our loss design:
>
> 1. Reference as Training Headroom: The model $\theta_\text{ref}$ serves as a local performance benchmark, capturing task-specific distributions that the proxy should respect.
>
> 2. Excess Loss Mechanism: When disagreements occur, $\mathcal{L}_\text{exc}$ ensures that the proxy model's training aligns with the reference's performance. This prevents the proxy from learning weights that would degrade local task performance.
>
> **Ablation Study Explanation:**
> We'd like to address the misunderstanding and the part we did not addressed enough in the main text.
>
> * **FedRAM** (Experiment 5):
>
> $\theta_\text{ref}$ →  $\theta_\text{proxy}$  → $\theta_\text{proxy}$ and $\theta_\text{ref}$
>
> $~~~~~~~~~~~$  |   $~~~~~~~~~~~~~~~~~~~~~~~~$    |
>
> $~~~~~~~~~~$  $\boldsymbol{w}$    $~~~~~~~~~~~~~~~~~~~~~~~$    $\boldsymbol{\alpha}$
>
> $~~~~~~~~~~$  └-----------------------└------→ $\theta_\text{agent}$ → Evaluation
>
> * **Experiment 1**: w/o $\theta_\text{ref}$, $\theta_\text{proxy}$, we train and evaluate on the FL agent model $\theta_\text{agent}$ (FedAvg). Also, $\boldsymbol{\alpha}$ and $\boldsymbol{w}$ are not involved, which equals to FedAvg and serves as the comparison baseline against Exp 6 and 7.
>
> $\quad\quad\quad~~~~~~~~~~~~~~~~~~~~~~~~~~~~~~~~~~~~~~~~$ $\theta_\text{agent}$ → Evaluation
>
> * **Experiment 2**: w/o $\theta_\text{ref}$, we train $\theta_\text{proxy}$ in FL without the use of excess loss $\ell_\text{exc}$ while keeping differential loss $\ell_\text{dif}$. In this setting, we evaluate $\theta_\text{agent}$ solely based on client weights $\eta_\text{client}$:
>
> $~~~~~~~~~~~$  $\theta_\text{proxy}$
>
> $~~~~~~~~~~~~$  |
>
> $~~~~~~~~~~~$  $\boldsymbol{w}$
>
> $~~~~~~~~~~~$  └-----------------------------→ $\theta_\text{agent}$ → Evaluation
>
> * **Experiment 3**: w/o $\theta_\text{proxy}$. In this setting, we can not derive the task weights $\boldsymbol{w}$ and client weights $\boldsymbol{\alpha}$, so that we directly evaluate on $\theta_\text{ref}$.
>
> $\theta_\text{ref}$ -----┐
>
> $~~~~~~~~~~~~$  |
>
> $~~~~~~~~~~~~$ └-----------------------------------------→ Evaluation
>
> * **Experiment 4**: w/o $\theta_\text{agent}$. We evaluate on $\theta_\text{proxy}$ in this case.
>
> $\theta_\text{ref}$ →  $\theta_\text{proxy}$  → $\theta_\text{proxy}$ and $\theta_\text{ref}$
>
> $~~~~~~~~~~~~$  |
>
> $~~~~~~~~~~~~$  └-----------------------------------------→ Evaluation
>
> Based on the original ablation study, we extend to two more experiments:
>
> * **Experiment 6**: w/o $\boldsymbol{\alpha}$. $\theta_\text{agent}$ is trained using client weights $\boldsymbol{w}$:
>
> $\theta_\text{ref}$ →  $\theta_\text{proxy}$
>
> $~~~~~~~~~~~~$  |
>
> $~~~~~~~~~~~$  $\boldsymbol{w}$
>
> $~~~~~~~~~~~$  └-----------------------------→ $\theta_\text{agent}$ → Evaluation
>
> * **Experiment 7**: w/o $\boldsymbol{w}$. $\theta_\text{agent}$ is trained using task weights $\boldsymbol{\alpha}$:
>
> $\theta_\text{ref}$ →  $\theta_\text{proxy}$  → $\theta_\text{proxy}$ and $\theta_\text{ref}$
>
> $~~~~~~~~~~~~~~~~~~~~~~~~~~~~~~~~~~~~~~$  |
>
> $~~~~~~~~~~~~~~~~~~~~~~~~~~~~~~~~~~~~~~$    $\boldsymbol{\alpha}$
>
> $~~~~~~~~~~~~~~~~~~~~~~~~~~~~~~~~~~~~~~$ └------→ $\theta_\text{agent}$ → Evaluation
>
> **Experimental Validation:**
> Our ablation studies (Section 5.2.4, Table 3) systematically validate the synergistic contributions of each component. The results demonstrate that:
>
> (1) **Reference model serves as the local training headroom**: By comparing **Experiment 1 and 3**, we compare the direct performance of $\theta_\text{agent}$ (In Exp. 1) and $\theta_\text{ref}$ (In Exp. 3). In general, while both $\theta_\text{ref}$ and $\theta_\text{agent}$ can generalize to global evaluation, $\theta_\text{ref}$ fail to do well on local evaluation for smaller model size than $\theta_\text{agent}$.
>
> (2) **Proxy model do not necessarily learn the global representations**:
> The performance on $\theta_\text{proxy}$ (Exp 4) suffers a great drop compared to Experiment 1. This is because proxy models are trained to tune $\boldsymbol{\alpha}$ and $\boldsymbol{w}$. The knowledge $\theta_\text{proxy}$ learned (tuning hyperparameters) fail to apply to the dataset.
>
> ### **W3: All experiments are based on T5-small/base models. While practical, it leaves questions on generality across modalities or architectures.**
>
> **A3:**
>
> **For extending to vision tasks**, we already presented the results in **Appendix B.6 (d)**. In this setting, we choose swin transformer to be our architecture for $\theta_\text{ref}$ and $\theta_\text{proxy}$, and clip-vit for $\theta_\text{agent}$.  We base our experiments on seven distinct CV datasets: MNIST, EuroSat, Fashion-MNIST, SVHN, DTD, Stanford-Cars, and CIFAR-10.
>
> **For architectures**, In Appendix B.6 (4) , we applied **Swin Transformer** and **CLIP-ViT**, which are not from the same model family. Based on the results in Appendix Table 9, we extend the experiments as follows:
>
> | Exp | Modality| $\theta_{ref}$ and $\theta_{proxy}$  | $\theta_{agent}$  | Global | Local | ID   | OOD  |
> |-|-|-|-|-|-|-|-|
> |1| **cv** |Swin-T | CLIP-ViT  | 82.05 | 79.23  | 70.94 | 65.26 |
> |2| **cv** |CLIP-ViT | Swin-T  | 79.39 | 80.59  | 73.52 | 62.67 |
> |3| nlp |T5-base | T5-large| 84.29   | 90.29  | 91.53 | 87.26 |
> |4| nlp |T5-large | LLama  | 86.77   | 92.03  | 92.24 | 88.19 |
> |5| nlp |bert | T5-large   | 84.80   | 90.41  | 91.64 | 85.36 |
> |6| nlp |bert | LLama   | 84.43   | 89.64  | 91.78 | 87.40 |
>
> To conclude, experiments both in CV (Exp 1, 2) and NLP (Exp 3~6) can validate FedRAM's generalization to model types.
>
>
> ### **W4: The choice of $\eta_{task}$, $\eta_{client}$, smoothing factor s and their effects are not discussed in the main text.**
>
> **A4:**
> 1. We employ a **logarithmic grid search**, ranging from 1E-2 to 1E2.  Our systematic analysis identified the optimal value as 1E-1, with the effective parameter range being $[-1, 1]$, as in ** Appendix B Figure 8**. Within this range, the impact on global framework performance remains consistently stable.
> 2. Take $\eta_\text{task}$ as an example. Theoretically, assuming $\theta_\text{ref}$ in Step 1 is well trained, and the loss $\ell^\tau_\text{ref}$ equals to 0. The term $\ell^\tau_\text{exc}$ equals to $\frac{\ell_{\text{proxy}}^\tau}{{|\mathcal{D}^\tau_L|}}$. Assuming $\ell^\tau_\text{proxy}$ goes down per iteration. We make sure the first-iteration of $\alpha^1_\tau\leftarrow\alpha^0_\tau e^{ \eta_\text{task} \ell^\tau_\text{exc} } \leq 1$ should not excess the natural upper bound 1 (note that $\alpha^0_\tau=\frac{1}{K}$). Therefore, it is less risky to set  $\eta_\text{task}\leq\frac{\text{ln}K}{\ell^\tau_\text{exc} }=\frac{|\mathcal{D}^\tau_L| \times\text{ln}K}{\ell^\tau_\text{proxy}}$, where $|\mathcal{D}^\tau_L|$ is denoted as the training dataset size., $K$($K>1$) as the number of clients, $\ell^\tau_\text{proxy}$ as the maximum value of first step proxy loss performed on task $\tau$.
> 3. We do not recommend extensive tuning for these parameters. For practical implementations, $\eta = 1, 0.1$ or $0.01$ would be simple and effective default values.
>
> ### **W5: FedRAM could be enhanced by including the areas of personalized federated learning and personalized device-cloud collaboration, if time allows.**
>
> **A5:**
>
> We appreciate the suggestion to include experiments in personalized federated learning (PFL) . However, we would like to clarify that FedRAM primarily focuses on global robustness and performance in **FL**, rather than personalization (**pFL**). We believe adding pFL comparisons may be **misaligned**, as the objectives and evaluation metrics differ significantly. To achieve the similar training goals, pFL would greatly harm the global convergence and tend to take up greater computational resources.

---

> > ### Comment · Reviewer_9ir2 · 2025-08-06
> > **Official comments by reviewer 9ir2**
> >
> > Thank you to the authors for the rebuttal, which addressed my concerns. Therefore, I will keep my score unchanged.

---

### Official Review · Reviewer_gkRF · 2025-07-03

**Clarity:** 3
**Significance:** 2
**Originality:** 2
**Rating:** 4
**Confidence:** 3

**Summary:**

FedRAM proposes a federated multi-task learning (FL-MTL) framework to address statistical heterogeneity, task interference, and global-local trade-offs in distributed settings. Its main contribution includes:

1. Propose a novel reference-proxy-agent strategy for decoupled optimization.
2. Adaptive tuning of the task importance parameter and client contribution parameter.
3. Experiments on 6 benchmarks show ​​the accuracy gains​​ in in-domain and out-of-domain tasks

**Questions:**

1. Is the assumption in Theorem 2 (Appendix A) about idealized weight updates ( $w_k \propto 1 / \sigma_k^2$ ) reasonable?
2. Add experiments measuring gradient variance across clients with/without proxy adjustments.
3. Include more recent FL-MTL baselines and report wall-clock time.
4. Add metrics like task-specific variance or min-client accuracy.

**Ethical Concerns:**

["NO or VERY MINOR ethics concerns only"]

**Final Justification:**

Many of the concerns have been addressed.

**Limitations:**

yes

**Paper Formatting Concerns:**

-

**Quality:**

2

**Strengths And Weaknesses:**

​​Strengths​​
1. Hierarchical model design decouples local adaptation, weight tuning, and global aggregation
2. The proposed proxy models reduce computation by guiding the agents

Weaknesses​
1. The baseline comparison is limited
2. The Ablation study (Table 3) omits component-wise impact
3. Experiments (Fig. 5) show severe task/data imbalance, but FedRAM's resampling lacks fairness guarantees for minority tasks.

---

> ### Author Rebuttal · Authors · 2025-07-31
>
> We greatly appreciate your thoughtful comments and constructive feedback, which have been invaluable in improving our manuscript. We have focused our responses on the following key issues:
>
> ---
>
> ### **W1: The Ablation study (Table 3) omits component-wise impact.**
>
> **A1:** **Ablation Study Explanation:**
> We'd like to address the misunderstanding and the part we did not addressed enough in the main text.
>
> * **FedRAM (Experiment 5)**:
>
> $\theta_\text{ref}$ →  $\theta_\text{proxy}$  → $\theta_\text{proxy}$ and $\theta_\text{ref}$
>
> $~~~~~~~~~~~$  |   $~~~~~~~~~~~~~~~~~~~~~~~~$    |
>
> $~~~~~~~~~~$  $\boldsymbol{w}$    $~~~~~~~~~~~~~~~~~~~~~~~$    $\boldsymbol{\alpha}$
>
> $~~~~~~~~~~$  └-----------------------└------→ $\theta_\text{agent}$ → Evaluation
>
> * **Experiment 1**: w/o $\theta_\text{ref}$, $\theta_\text{proxy}$, we train and evaluate on the FL agent model $\theta_\text{agent}$ (FedAvg). Also, $\boldsymbol{\alpha}$ and $\boldsymbol{w}$ are not involved, which equals to FedAvg and serves as the comparison baseline against Exp 6 and 7.
>
> $\quad\quad\quad~~~~~~~~~~~~~~~~~~~~~~~~~~~~~~~~~~~~~~~~$ $\theta_\text{agent}$ → Evaluation
>
> * **Experiment 2**: w/o $\theta_\text{ref}$, we train $\theta_\text{proxy}$ in FL without the use of excess loss $\ell_\text{exc}$ while keeping differential loss $\ell_\text{dif}$. In this setting, we evaluate $\theta_\text{agent}$ solely based on client weights $\eta_\text{client}$:
>
> $~~~~~~~~~~~$  $\theta_\text{proxy}$
>
> $~~~~~~~~~~~~$  |
>
> $~~~~~~~~~~~$  $\boldsymbol{w}$
>
> $~~~~~~~~~~~$  └-----------------------------→ $\theta_\text{agent}$ → Evaluation
>
> * **Experiment 3**: w/o $\theta_\text{proxy}$. In this setting, we can not derive the task weights $\boldsymbol{w}$ and client weights $\boldsymbol{\alpha}$, so that we directly evaluate on $\theta_\text{ref}$.
>
> $\theta_\text{ref}$ -----┐
>
> $~~~~~~~~~~~~$  |
>
> $~~~~~~~~~~~~$ └-----------------------------------------→ Evaluation
>
> * **Experiment 4**: w/o $\theta_\text{agent}$. We evaluate on $\theta_\text{proxy}$ in this case.
>
> $\theta_\text{ref}$ →  $\theta_\text{proxy}$  → $\theta_\text{proxy}$ and $\theta_\text{ref}$
>
> $~~~~~~~~~~~~$  |
>
> $~~~~~~~~~~~~$  └-----------------------------------------→ Evaluation
>
> Component wise, to better assess the results, please refer to Exp 1, 3 and 4.
>
> * **Experiment 6**: w/o $\boldsymbol{\alpha}$. $\theta_\text{agent}$ is trained using client weights $\boldsymbol{w}$:
>
> $\theta_\text{ref}$ →  $\theta_\text{proxy}$
>
> $~~~~~~~~~~~~$  |
>
> $~~~~~~~~~~~$  $\boldsymbol{w}$
>
> $~~~~~~~~~~~$  └-----------------------------→ $\theta_\text{agent}$ → Evaluation
>
> * **Experiment 7**: w/o $\boldsymbol{w}$. $\theta_\text{agent}$ is trained using task weights $\boldsymbol{\alpha}$:
>
> $\theta_\text{ref}$ →  $\theta_\text{proxy}$  → $\theta_\text{proxy}$ and $\theta_\text{ref}$
>
> $~~~~~~~~~~~~~~~~~~~~~~~~~~~~~~~~~~~~~~$  |
>
> $~~~~~~~~~~~~~~~~~~~~~~~~~~~~~~~~~~~~~~$    $\boldsymbol{\alpha}$
>
> $~~~~~~~~~~~~~~~~~~~~~~~~~~~~~~~~~~~~~~$ └------→ $\theta_\text{agent}$ → Evaluation
>
> When it comes to component-wise ablation study, we compare Exp 1, 3, and 4, and we evaluate $\theta_\text{agent}$, $\theta_\text{ref}$, $\theta_\text{proxy}$, respectively. It is observed that:
> 1. $\theta_\text{agent}$ outperforms the other components both in global and local performance, for its larger model size and trainable parameters.
> 2. $\theta_\text{ref}$ has a competitive performance on global evaluation but fail to improve on the local performance compared to $\theta_\text{agent}$.
> 3. $\theta_\text{proxy}$ shows a giant gap among the components, because it does not necessarily learn the local and global knowledge in this system but to tune hyperparamters $\boldsymbol{\alpha}$ and $\boldsymbol{w}$.
>
> | Exp  | Method | $\quad$ | Evaluation on | Global | Local  |
> |---------|--------|---------|--------|-------|------|
> | 1 | w/o $\theta_\text{ref}$, $\theta_\text{proxy}$ | | $\theta_\text{agent}$ | 72.71   | 76.34  |
> | 2 | w/o $\theta_\text{ref}$ |  | $\theta_\text{agent}$ | 71.17   | 73.58 |
> | 3 | w/o $\theta_\text{proxy}$ |  | $\theta_\text{ref}$  | 72.54   | 70.52  |
> | 4 | w/o $\theta_\text{agent}$ |  | $\theta_\text{proxy}$     | 2.69   | 2.51  |
> | 5 | FedRAM |  |  $\theta_\text{agent}$     | 75.94   | 79.62  |
>
> **Ablation Study on Hyperparameters**: Based on the original ablation study, we extend to two more experiments on the hyperparameters' impact. Exp 6 and 7 both show improvements uppon Exp 1, demonstrating their effectiveness.
>
> | Exp  | Method  | Weights | Framework $\quad\quad\quad$ | Evaluation on | Global | Local  |
> |---------|---------|--------|-------|------|------|------|
> | 1 | w/o $\boldsymbol{\alpha}$, $\boldsymbol{w}$ | None  | $\theta_\text{ref}$, $\theta_\text{proxy}$, $\theta_\text{agent}$ | $\theta_\text{agent}$ | 72.71   | 76.34  |
> | 6 | w/o $\boldsymbol{\alpha}$  | $\boldsymbol{w}$ | $\theta_\text{ref}$, $\theta_\text{proxy}$, $\theta_\text{agent}$ | $\theta_\text{agent}$ | 73.57   |  79.51  |
> | 7 | w/o $\boldsymbol{w}$    | $\boldsymbol{\alpha}$ | $\theta_\text{ref}$, $\theta_\text{proxy}$, $\theta_\text{agent}$  |$\theta_\text{agent}$  | 75.20   | 76.78  |
> | 8 | FedRAM | $\boldsymbol{\alpha}$, $\boldsymbol{w}$ | $\theta_\text{ref}$, $\theta_\text{proxy}$, $\theta_\text{agent}$ | $\theta_\text{agent}$     | 75.94   | 79.62  |
>
> ### **W2: Experiments (Fig. 5) show severe task/data imbalance, but FedRAM's resampling lacks fairness guarantees for minority tasks.**
>
> **A2:** Task and data imbalance is indeed an inherent challenge in federated multi-task learning under non-IID conditions.
>
> Our resampling strategy inherently protects minority tasks through several key mechanisms:
>
> 1. Task Weight ($\boldsymbol{\alpha}$): The proxy model learns individual importance weights $\alpha_\tau$ for each task $\tau$, ensuring that minority tasks receive appropriate representation regardless of their data volume. As demonstrated in **Figure 7**, our task weights dynamically adjust from the initial data size distribution (**red line**) to more balanced configurations (**blue lines**). Notably, when $\eta$=0.1, minority tasks like PAWS and WikiQA receive significantly increased weights compared to their original data proportions.
>
> 2. Distributionally Robust Optimization: The resampling mechanism employs principles similar to group distributionally robust optimization [1], where the local training process of proxy model balances performance across all tasks. As demonstrated in Figure 7, all task weights are distributed within the range of [0.2, 0.4] compared to the dataset size range [0, 0.8].
>
> [1] Sagawa et al. Distributionally robust neural networks for group shifts. ICLR 2020.
>
>
> ### **Q1: Is the assumption in Theorem 2 (Appendix A) about idealized weight updates ( $\boldsymbol{w}_k \propto 1/\sigma_k^2$) reasonable?**
>
> **Response to Q1:** The assumption $w_k \propto 1/\sigma_k^2$ in Theorem 2 is practically reasonable for **Fairness**: FL clients with high $\sigma^2$ (caused by noise or extremely heterogeneous data) can harm model convergence or suppress the contributions of the minorities if weighted equally. Downweighting them (as in the assumption) is to achieve worst-case robustness by prioritizing clients with more reliable data. In this case, dynamically **upweights high-loss data groups** during training to improve worst-group performance. Groups with higher loss variance (or higher loss) are upweighted to ensure robustness.
>
> ### **Q2: Add experiments measuring gradient variance across clients with/without proxy adjustments.**
>
> **Response to Q2:**
> Setup: The experiment setup is aligned with **Section 5.1**.
> Implementation: Track variance metrics with/without proxy adjustments at communication rounds 1, 10, 20, 30 and 40.
>
> | Method  | proxy adjustments $\quad$ |  Round 1  |  Round 10 |  Round 20 |  Round 30 |  Round 40
> |---------|--------|-------|------|------|------|------|
> | FedAvg | No  | 4.33e6   | 8.46e5   | 1.13e4   | 8.72e2   |   2.26e-3   |
> | FedRAM | yes | 2.67e6   | 3.58e4   | 9.67e3   | 4.48e2   |   1.86e-3   |
>
> ### **Q3: Include more recent FL-MTL baselines and report wall-clock time.**
>
> **Response to Q3:**
>
> Thank you for the suggestion. As shown in Section 5.2.3 **Table 2**, we have already presented wall-clock time (Computational Cost) for all methods. Our FedRAM achieves the fastest training time of 12,060 seconds, which is 14.2% faster than FedAvg (the second).
>
> We add more experiments here:
>
> | Methods | Global |  Local  |  ID |  OOD |  Computational Cost (s) |
> |---------|--------|-------|------|------|------|
> | pFedFDA [1] | 74.12 | 77.38  | 71.77  | 75.68  |  23603  |
> | OBF [2] | 69.22   | 	75.75   | 70.93   | 67.48    |  32497  |
> |FedHCA$^2$ [3] | 71.31   |  80.56	  | 72.14   |  70.45   | 42506  |
> | **FedRAM** (ours) | 75.94   |  79.62  |  72.89  |  76.32  |   12060  |
>
> [1]. Mclaughlin, et al. "Personalized federated learning via feature distribution adaptation." NIPS, 2024.
> [2].Vani, Ankit, et al. "Forget Sharpness: Perturbed Forgetting of Model Biases Within SAM Dynamics." ICML. 2024.
> [3]. Lu, Yuxiang, et al. "Fedhca2: Towards hetero-client federated multi-task learning." CVPR/CVF, 2024.
>
> ### **Q4: Add metrics like task-specific variance or min-client accuracy.**
>
> **Response to Q4:**
>
> Thank you for the suggestion. As shown in main text **Table 1**, we have already presented min-client accuracy (shown as **Bottom Decile** in the table) for all methods. We will update a clearer description about the Bottom Decile.
>
> We add task-specific variance of the Global F1-score metric here:
>
> | Methods | min-client accuracy |  task-specific variance  |
> |---------|--------|-------|
> | FedAvg | 54.79 |  1.27  |
> | FedProx |55.45  | 	0.93   |
> | Ditto |  62.37   | 0.56  |
> | Fedrep | 66.89 | 0.33  |
> | MOON |  56.30   | 0.78  |
> | DBE | 58.53   |  1.12	  |
> | FedMTL | 62.32   |   0.82	  |
> | FedBone | 61.60   |   0.50	  |
> | **FedRAM** (ours) |  73.21   | 0.24  |

---

> > ### Author Response · Authors · 2025-08-07
> >
> > Dear reviewer gkRF,
> >
> > We hope that our rebuttal has sufficiently addressed your concerns. If you have any questions or items to discuss, please let us know as the rebuttal period is about to end really soon.

---

> > ### Comment · Area_Chair_MSnU · 2025-08-07
> >
> > Dear Reviewer gkRF,
> >
> > Can you please read the author's response and get back to them asap?
> > We are asked to report non-participating reviewers this year. Please submit your review response ASAP. I really appreciate your help.
> >
> >
> > Thanks,
> >
> > AC

---

### Official Review · Reviewer_dqPQ · 2025-07-03

**Clarity:** 3
**Significance:** 3
**Originality:** 3
**Rating:** 5
**Confidence:** 5

**Summary:**

This paper proposes FedRAM, a novel federated learning framework for multi-task learning (FL-MTL). FedRAM introduces a three-step framework that updates two scalar hyperparameters: task importance weights and client aggregation coefficients. It employs a reference-proxy-agent strategy, where the proxy model serves as an intermediate between the local reference model and the global agent model. The framework aims to improve performance on both in-domain and out-of-domain tasks while reducing computational costs.

**Questions:**

1. What task was used to optimize θref in step 1? Is it one of the multi-tasks?

2. Is there a clear guideline on selecting ηtask and ηclient beyond grid search?

**Ethical Concerns:**

["NO or VERY MINOR ethics concerns only"]

**Final Justification:**

I read the rebuttal, and I will raise my score accordingly.

**Limitations:**

Yes

**Quality:**

3

**Strengths And Weaknesses:**

Strengths:

1.	FedRAM introduces a novel three-step framework that effectively addresses the challenges of statistical heterogeneity and task interference in FL-MTL.

2.	The use of excess loss to tune task weights and differential loss to adjust client weights is principled and empirically effective.

3.	Convergence proofs are provided and align with standard assumptions in federated learning literature.

Weaknesses:

1.	The three-step framework and the need to train multiple models (reference, proxy, and agent) may add complexity to the implementation process.

2.	Performance depends on exponential scaling factors and smoothing parameters, but these are only lightly addressed in the main text.

---

> ### Author Rebuttal · Authors · 2025-07-31
>
> Thank you for your insightful feedback and helpful suggestions. dqPQ notes that “FedRAM introduces a novel three-step framework” with "principled and empirically effective loss use", and “Convergence proofs are provided and align with standard assumptions”. We address specific questions below:
>
> ---
>
> ### **W1: The three-step framework and the need to train multiple models (reference, proxy, and agent) may add complexity to the implementation process.**
>
> **A1:** We acknowledge that our three-stage framework introduces additional complexity compared to traditional federated learning approaches. However, this design choice provides significant flexibility by maintaining compatibility with existing FL methods, allowing practitioners to integrate FedRAM components selectively based on their specific requirements. The apparent complexity stems from our deliberate decoupling of three critical optimization objectives in FL-MTL scenarios: task weights ($\boldsymbol{\alpha}$), client weight distribution ($\boldsymbol{w}$), and model parameter optimization ($\theta_\text{agent}$). This decomposition strategy substantially reduces computational overhead, as our reference and proxy models are approximately 1/25 the size of the agent model, contributing only ~15% of the total training time. Furthermore, in practical deployment scenarios, organizations often possess pre-trained smaller models already adapted to local datasets, which can serve directly as reference models. In such cases, the primary training effort focuses on the proxy and agent models.
>
> ---
>
> ### **W2: Performance depends on exponential scaling factors and smoothing parameters, but these are only lightly addressed in the main text.**
>
> ### **Q2: Is there a clear guideline on selecting $\eta_{task}$ and $\eta_{client}$ beyond grid search?**
> **A2:** We appreciate the reviewer's concern regarding hyperparameter sensitivity. For **smoothing parameters**, we consistently use 0.01 across all experiments, chosen to balance convergence stability without significantly increasing communication rounds or overly influencing weight dynamics. This value ensures that both task weights $\boldsymbol{\alpha}$ and client weights $\boldsymbol{w}$) converge within reasonable bound while maintaining stability.
>
> Regarding **exponential scaling factors** $\eta_\text{task}$ and $\eta_\text{client}$, while the main text prioritizes the core algorithmic contributions of our three-step architecture and the sequential solutions for $\boldsymbol{\alpha}$, $\boldsymbol{w}$ and $\theta_{agent}$, we provide $\eta_\text{task}$ and $\eta_\text{client}$ analysis in Appendix B.3.  We'd like to add how we find the $\eta_\text{task}$ and $\eta_\text{client}$ here:
>
> 1. We employed a **logarithmic grid search strategy**, ranging from 1E-2 to 1E2.  The optimal logarithmic value is 1E-1, with the effective parameter range being $[-1, 1]$, as demonstrated in **Figure 8**. Within this range, the impact on global performance remains consistently stable.
>
> 2. Take $\eta_\text{task}$ as an example. Theoretically, assuming $\theta_\text{ref}$ in Step 1 is well trained, and the loss $\ell^\tau_\text{ref}$ converges to 0. The term $\ell^\tau_\text{exc}$ equals to $\frac{\ell_{\text{proxy}}^\tau}{{|\mathcal{D}^\tau_L|}}$. Assuming $\ell^\tau_\text{proxy}$ will effectively go down per iteration as is trained. We make sure the first-iteration of $\alpha^1_\tau\leftarrow\alpha^0_\tau e^{ \eta_\text{task} \ell^\tau_\text{exc} } \leq 1$ should not excess the natural upper bound 1 (note that $\alpha^0_\tau=\frac{1}{K}$). Therefore, it is less risky to set  $\eta_\text{task}\leq\frac{\text{ln}K}{\ell^\tau_\text{exc} }=\frac{|\mathcal{D}^\tau_L| \times\text{ln}K}{\ell^\tau_\text{proxy}}$, where $|\mathcal{D}^\tau_L|$ is denoted as the training dataset size., $K$($K>1$) as the number of clients, $\ell^\tau_\text{proxy}$ as the maximum value of first step proxy loss performed on task $\tau$.
>
> 3. We do not recommend extensive tuning for these parameters. For practical implementations, $\eta = 1, 0.1$ or $0.01$ can be simple and effective default values.
>
> ### **Q1: What task was used to optimize $\theta_\text{ref}$ in step 1? Is it one of the multi-tasks?**
>
> **Response to Q1:**
> The reference model is trained on all multi-tasks assigned to each client, using their local training data without any task weighting. During Step 1, each client performs standard multi-task learning with equal task importance, ensuring the reference model captures baseline local knowledge across all tasks.
>
> The key difference from subsequent stages is that while the reference model uses uniform task treatment, both proxy and agent models incorporate learned task weights $\boldsymbol{\alpha}$ to dynamically adjust task sampling ratios during training. The task categories remain identical across all three steps—only the dataset sizes are changing based on computed task weights and resampling.

---

> > ### Comment · Reviewer_dqPQ · 2025-08-05
> >
> > I read the rebuttal, and I will raise my score accordingly.

---

> > > ### Author Response · Authors · 2025-08-06
> > > **Thank You**
> > >
> > > Dear Reviewer dqPQ
> > >
> > > Thank you very much for your thoughtful and constructive feedback. We truly appreciate your time and effort in reviewing our work.
> > >
> > > Best Regards,
> > >
> > > Authors of Paper 6648

---

### Official Review · Reviewer_m3Lj · 2025-07-23

**Clarity:** 3
**Significance:** 3
**Originality:** 2
**Rating:** 5
**Confidence:** 4

**Summary:**

This paper introduces a novel federated multi-task learning framework, FedRAM, which uses a proxy model to balance client contributions. The proposed framework shows at least 3% improvement in accuracy across metrics on diverse NLP benchmarks. It also shows up to 15X reduction in computational cost.

**Questions:**

* Although the paper is generally easy to follow, some of the notations can be simplified and cleaner
   * For example, around line 100, what's the point of introducing $\theta_1$, $\theta_2$? They bring unnecessary confusions. Around line 106, $\theta^+$ and $\theta^*$ are defined but almost never used until very end of the paper. If the authors have to show the size of the model, why not directly use something like $\theta_{\text{ref}}^{+}$.
* Regarding the rigorousness
    * For example, around line 124, do we assume the loss is just MSE? In eq. (6), $\ell_{\text{proxy}}^{\tau}$ and $\ell_{\text{ref}}^{\tau}$  are not defined. Also I guess the notations here are also ambiguous since the losses are missing indices $k$ representing clients. In step 3 of Algorithm 1: do we just perform one step of gradient descent update? In line 14 of Algorithm 1, should the second term of the product be  $e^{\eta_\text{client}\ell_\text{dif}}$. Mixed use of $\mathcal{L}$ and $\ell$.

* Do we really need the proxy model? Can we just train the agent model by comparing with the reference and dynamically adjust $\alpha$ and $w$. I see the regression on performance in the ablation study section. However I wonder the detailed setup of the ablation experiments. Maybe I missed them?
* The authors discuss the sensitivities on the hyper-parameter $\eta_{\text{client}}$ and $\eta_{\text{task}}$. What is the hyper-parameter generalization? What kind of approach the authors use to find the optimal ones. For example, if we apply the framework to another set of tasks, how to effectively find the optimal hyper-parameter? Could the authors discuss the effort finding them?
* The balances of client contributions heavily depend on the quality of reference models. The authors use T5 family models for the small reference and large agent models. How sensitive the framework is with respect to the sizes and choices of reference model? Do we need them to be in the same family to have desired results. It would be more convincing to see performances of various pairs of reference and agent models.

In summary, I think in general the paper is good. Since the proxy model idea has been around, the novelty of the paper is relatively limited. I think the authors can extend their experiments a bit, in particular on the choices of reference mode. I am happy and willing to raise my scores if the authors can address mainly the last two question.

---

After reading the rebuttal from the authors, my questions and concerns have been addressed. As the authors promise to incorporate the rebuttal into their camera-ready version, I raise my score from 3 to 5. Thanks again the authors for their detailed responses.

**Ethical Concerns:**

["NO or VERY MINOR ethics concerns only"]

**Final Justification:**

After reading the rebuttal from the authors, my questions and concerns have been addressed. As the authors promise to incorporate the rebuttal into their camera-ready version, I raise my score from 3 to 5. Thanks again the authors for their detailed responses.

**Limitations:**

See above.

**Paper Formatting Concerns:**

See above.

**Quality:**

3

**Strengths And Weaknesses:**

### Quality:
- The paper is generally well-written and easy to follow. However, in some paragraphs, the presentation, notations and the rigorousness could be better.

### Clarity:
- The paper is mostly clear.

### Significance:
- The framework is straightforward and can be easily applied to any federated multi-task learning tasks.
- Experiments show the non-trivial performance improvement while using the least computational power compared with methods in the literature.

### Originality:
- The idea of using proxy models to evaluate the difficulties of different tasks is not new in multi-task learning but it seems that it is novel to apply it on the federated learning.

---

> ### Author Rebuttal · Authors · 2025-07-31
>
> We sincerely thank you for the insightful comments and valuable suggestions, which have greatly contributed to enhancing our manuscript. Based on the given weaknesses and questions, the key issues we address in our response include:
>
> ---
>
> ### **Q1: Some of the notations can be simplified and cleaner**
> 1. Around line 100, what's the point of introducing $\theta_1$ and $\theta_2$?
> 2. $\theta^+$ and $\theta^*$ are defined but almost never used until very end of the paper
>
> **A1:** Thanks for pointing out. We will carefully adjust the use of notations.
>
> ---
>
> ### **Q2: Regarding the rigorousness:**
> 1. Around line 124, do we assume the loss is just MSE?
> 2. In eq. (6), $\ell_{\text{ref}}^\tau$ and $\ell_{\text{proxy}}^\tau$ are not defined. Also  the notations here are also ambiguous since the losses are missing indices $k$ representing clients.
> 3. In step 3 of Algorithm 1: do we just perform one step of gradient descent update?
> 4. In line 14 of Algorithm 1, should the second term of the product be $e^{\eta_\text{client}\ell_\text{dif}}$. Mixed use of $\mathcal{L}$ and $\ell$.
>
> **A2:**
> 1. The loss function should be more flexible and will be replaced with $\min_{\theta^L_\text{ref}} \mathcal{L}(\theta^L_\text{ref}) = \frac{1}{|D_L|} \sum_{(x,y) \in D_L} \ell(f_{\theta^L_\text{ref}}(x), y)$
> 2. We use $\ell^\tau = \mathcal{L}(\mathcal{D}^\tau_{L};\theta^L)$ to present task-wise loss. Thanks for pointing out the typo.
> 3. No, we don't. Step 3 is a complete local training period, which consists of multiple gradient descent updates.
> 4. Thanks for pointing out the typo. We will kindly adjust the use of $\mathcal{L}$ and $\ell$.
>
>
> ---
>
> ### **Q3: Do we really need the proxy model?**
> 1. Can we just train the agent model by comparing with the reference and dynamically adjust $\boldsymbol{\alpha}$ and $\boldsymbol{w}$?
>
> 2. The detailed setup of the ablation experiments.
>
>
> **A3:** Yes, we do need the proxy model!
>
> 1.**No, we cannot.** In this response, we will discuss your case first and then summarize the proxy model's functions.
>
> **In your pre-assumed case**, removing proxy model will cause two major problems:
> * **Non-IID Struggle**: Jointly optimizing [$\boldsymbol{\alpha}$, $\boldsymbol{w}$ and $\theta_{agent}$] makes it extremely **difficult to converge** to a stable performance, since it's continuing minimax process for independant training goals on $\boldsymbol{\alpha}$, $\boldsymbol{w}$ and $\theta_{agent}$, especially in federated non-IID settings.
> * **High Computation (in your pre-assumed case)**: Besides, training $\theta_{agent}$ by comparing with $\theta_{ref}$ **cannot disentangle the contributions** from multiple training goals. For instance, when client 1 shows a larger performance gap compared to all other clients, it remains unclear whether this stems directly from $\boldsymbol{w}$ or $\boldsymbol{\alpha}$. The cost of re-adjusting $\boldsymbol{w}$ and $\boldsymbol{\alpha}$ to make another FL training process is overwhelming.
>
> **Our proxy model**:
>
> * **Provides sequential solutions**: solve for model-agnostic hyperparameters [$\boldsymbol{\alpha}$, $\boldsymbol{w}$] first, and then train $\theta^*_{agent}$ in FL.
> * **Adaptation to Non-IID**: the locally trained $\theta^L_\text{proxy}$ learns prior knowledge of task heterogeneity; FL aggregation facilitates $\theta^G_\text{proxy}$ to share global knowledge.
> * **Low Computation**: according to Table 5 in our paper, the $\theta_\text{ref}$, $\theta_\text{proxy}$, and $\theta_\text{agent}$ exhibit the computational proportion of around 1%, 14%, 85% (the number of clients is set to 10), so that saving compute in training $\theta_{proxy}$ only results in a small compute benefit.
> * **Ablation Study** Our ablation study validates the effectiveness of $\theta_{proxy}$. We present the detailed setup and results of ablation studies below.
>
> 2.**The detailed setup of ablation studies** follow the same setup as in section 5.2.1, maintaining same datasets and client configurations.
> * **Experiment 5** **FedRAM**:
>
> $\theta_\text{ref}$ →  $\theta_\text{proxy}$  → $\theta_\text{proxy}$ and $\theta_\text{ref}$
>
> $~~~~~~~~~~~$  |   $~~~~~~~~~~~~~~~~~~~~~~~~$    |
>
> $~~~~~~~~~~$  $\boldsymbol{w}$    $~~~~~~~~~~~~~~~~~~~~~~~$    $\boldsymbol{\alpha}$
>
> $~~~~~~~~~~$  └-----------------------└------→ $\theta_\text{agent}$ → Evaluation
>
> * **Experiment 1**: w/o $\theta_\text{ref}$, $\theta_\text{proxy}$, we train and evaluate on the FL agent model $\theta_\text{agent}$ (FedAvg). Also, $\boldsymbol{\alpha}$ and $\boldsymbol{w}$ are not involved, which equals to FedAvg and serves as the comparison baseline against Exp 6 and 7.
>
> $\quad\quad\quad~~~~~~~~~~~~~~~~~~~~~~~~~~~~~~~~~~~~~~~~$ $\theta_\text{agent}$ → Evaluation
>
> * **Experiment 2**: w/o $\theta_\text{ref}$, we train $\theta_\text{proxy}$ in FL without the use of excess loss $\ell_\text{exc}$ while keeping differential loss $\ell_\text{dif}$. In this setting, we evaluate $\theta_\text{agent}$ solely based on client weights $\eta_\text{client}$:
>
> $~~~~~~~~~~~$  $\theta_\text{proxy}$
>
> $~~~~~~~~~~~~$  |
>
> $~~~~~~~~~~~$  $\boldsymbol{w}$
>
> $~~~~~~~~~~~$  └-----------------------------→ $\theta_\text{agent}$ → Evaluation
>
>
> * **Experiment 3**: w/o $\theta_\text{proxy}$. In this setting, we can not derive the task weights $\boldsymbol{w}$ and client weights $\boldsymbol{\alpha}$, so that we directly evaluate on $\theta_\text{ref}$.
>
> $\theta_\text{ref}$ -----┐
>
> $~~~~~~~~~~~~$  |
>
> $~~~~~~~~~~~~$ └-----------------------------------------→ Evaluation
>
> * **Experiment 4**: w/o $\theta_\text{agent}$. We evaluate on $\theta_\text{proxy}$ in this case.
>
> $\theta_\text{ref}$ →  $\theta_\text{proxy}$  → $\theta_\text{proxy}$ and $\theta_\text{ref}$
>
> $~~~~~~~~~~~~$  |
>
> $~~~~~~~~~~~~$  └-----------------------------------------→ Evaluation
>
> When it comes to component-wise ablation study, we compare Exp 1, 3, and 4. We evaluate $\theta_\text{agent}$, $\theta_\text{ref}$, $\theta_\text{proxy}$, respectively:
> 1. $\theta_\text{agent}$ outperforms the other components both in global and local performance, for its larger model size and trainable parameters.
> 2. $\theta_\text{ref}$ has a competitive performance on global evaluation but fail to improve on the local performance compared to $\theta_\text{agent}$.
> 3. $\theta_\text{proxy}$ shows a giant gap among the components, because it does not necessarily learn the local and global knowledge in this system but to tune hyperparamters $\boldsymbol{\alpha}$ and $\boldsymbol{w}$.
>
> | Exp  | Method  | Evaluation on | Global | Local  |
> |---------|---------|--------|-------|------|
> | 1 | w/o $\theta_\text{ref}$, $\theta_\text{proxy}$ | $\theta_\text{agent}$ | 72.71   | 76.34  |
> | 2 | w/o $\theta_\text{ref}$ | $\theta_\text{agent}$ | 71.17   | 73.58 |
> | 3 | w/o $\theta_\text{proxy}$ | $\theta_\text{ref}$  | 72.54   | 70.52  |
> | 4 | w/o $\theta_\text{agent}$ | $\theta_\text{proxy}$     | 2.69   | 2.51  |
> | 5 | FedRAM | $\theta_\text{agent}$     | 75.94   | 79.62  |
>
>
> ---
>
> ### **Q4: Hyper-paramters $\eta_\text{client}$ and $\eta_\text{task}$ Generalization**
> 1. Approach to find the optimal ones and the effort of finding them?
>
> 2.What is the hyper-parameter generalization?
>
> 3. How to effectively find the optimal Hyperparamters in another set of tasks?
>
> **A4:**
> 1. We employed a **logarithmic grid search** strategy, ranging from 1E-2 to 1E2.  The optimal logarithmic value in our framework is 1E-1, with the effective range being $[-1, 1]$, as in Figure 8. In this range, the impact on global performance remains consistently stable.
> 2. Take $\eta_\text{task}$ as an example. Theoretically, assuming $\theta_\text{ref}$ in Step 1 is well trained, and the loss $\ell^\tau_\text{ref}$ equals to 0. The term $\ell^\tau_\text{exc}$ equals to $\frac{\ell_{\text{proxy}}^\tau}{{|\mathcal{D}^\tau_L|}}$. Assuming $\ell^\tau_\text{proxy}$ goes down per iteration. We make sure the first-iteration of $\alpha^1_\tau\leftarrow\alpha^0_\tau e^{ \eta_\text{task} \ell^\tau_\text{exc} } \leq 1$ should not excess the natural upper bound 1 (note that $\alpha^0_\tau=\frac{1}{K}$). Therefore, it is less risky to set  $\eta_\text{task}\leq\frac{\text{ln}K}{\ell^\tau_\text{exc} }=\frac{|\mathcal{D}^\tau_L| \times\text{ln}K}{\ell^\tau_\text{proxy}}$, where $|\mathcal{D}^\tau_L|$ is denoted as the training dataset size., $K$($K>1$) as the number of clients, $\ell^\tau_\text{proxy}$ as the maximum value of first step proxy loss performed on task $\tau$.
> 3. We do not recommend extensive tuning for these parameters. For practical implementations, $\eta = 1, 0.1$ or $0.01$ can be simple and effective default values.
>
> ---
>
> ### **Q5: Model Types**
> 1.How sensitive is FedRAM with respect to the sizes and choices of $\theta_{ref}$?
>
> 2. Same family models? Performances of various pairs of $\theta_{ref}$ and $\theta_{agent}$.
>
> **A5:**
> 1. We discussed Agent-to-Proxy (A/P) Ratio in Appendix B.4. For a fixed size of $\theta_{agent}$, the size of $\theta_{ref}$ (along with $\theta_{proxy}$) can be 1/25 $\times$ smaller model scales.
> 2. No, we don't. In Appendix B.6 Task Scalability, we applied **Swin Transformer** as reference and proxy models, while **CLIP-ViT** as the agent models. Based on the results in Table 9, we extend the experiments to other model pairs as follows:
>
> | Exp | Modality| $\theta_{ref}$ and $\theta_{proxy}$  | $\theta_{agent}$  | Global | Local | ID   | OOD  |
> |-|-|-|-|-|-|-|-|
> |1| **cv** |Swin Transformer | CLIP-ViT  | 82.05 | 79.23  | 70.94 | 65.26 |
> |2| **cv** |CLIP-ViT | Swin Transformer  | 79.39 | 80.59  | 73.52 | 62.67 |
> |3| nlp |T5-base | T5-large| 84.29   | 90.29  | 91.53 | 87.26 |
> |4| nlp |T5-large | LLama  | 86.77   | 92.03  | 92.24 | 88.19 |
> |5| nlp |bert | T5-large   | 84.80   | 90.41  | 91.64 | 85.36 |
> |6| nlp |bert | LLama   | 84.43   | 89.64  | 91.78 | 87.40 |
>
> To conclude, experiments both in CV (Exp 1, 2) and NLP (Exp 3~6) can validate FedRAM's generalization to model types.

---

> > ### Comment · Reviewer_m3Lj · 2025-08-06
> >
> > I would like to thank the authors for their detailed and prompt responses. They mostly addressed my concerns and questions. Therefore, I will raise my score to 5 and recommend the paper to be accepted if the authors agree to incorporate the rebuttal in their final camera-ready version.

---

> > > ### Author Response · Authors · 2025-08-06
> > > **Thank You**
> > >
> > > Dear Reviewer m3Lj,
> > >
> > > Thank you for your positive feedback and constructive review. We’re glad our responses addressed your concerns, and we’ll incorporate the rebuttal into the final camera-ready version as suggested.
> > >
> > > Best Regards,
> > >
> > > Authors of Paper 6648

---

> > > ### Author Response · Authors · 2025-08-07
> > > **Follow Up**
> > >
> > > Dear Reviewer m3Lj,
> > >
> > > We truly appreciate your feedback and mentioning raising the score from 3 to 5 in your last comments.
> > >
> > > We wanted to follow up as we haven't yet received the mandatory acknowledgment.
> > >
> > > Please let us know if there's anything we can assist with.
> > >
> > > Warm Regards,
> > >
> > > Authors of Paper 6648

---

> > > > ### Comment · Reviewer_m3Lj · 2025-08-08
> > > >
> > > > Thank you for the follow up. The rebuttal solved my major concerns on the sensitivity of hyper-parameters and the choices of reference models. I have raised my score to 5. Thanks again for the detailed responses.

---

### Note · Authors · 2025-08-11

Dear AC, Reviewers:

We sincerely thank you and the reviewers for the constructive feedback during the rebuttal period. To facilitate your decision-making, we summarize the three major concerns raised and our comprehensive responses:

---

**Concern 1: Hyperparameters $\eta_\text{client}$ and $\eta_\text{task}$** -- From Reviewer m3Lj, dqPQ, 7drK

**Our Response**:

1.We employed a systematic logarithmic grid search (1E-2 to 1E2) identifying the optimal range [-1,1]. Our experimental validation in the paper Figure 8 (Appendix B.3) confirms parameter stability within this range.

2.Provided theoretical bounds: $\eta_\text{task}\leq\frac{\text{ln}K}{\ell^\tau_\text{exc} }=\frac{|\mathcal{D}^\tau_L| \times\text{ln}K}{\ell^\tau_\text{proxy}}$ for parameter setting.

**Reference**: Please refer to the results in “A4” for Reviewer m3Lj.

---

**Concern 2: Model Scalability** -- From Reviewer m3Lj, 7drK, 9ir2

**Our Response:** We conducted comprehensive scalability experiments and provided the comparison results.

**Reference**:

1. For **scalability to larger/backbone models**, please refer to “A1” for Reviewer 7drK.

2. For **scalability to models from different families**, please refer to “A5” for Reviewer m3Lj and “A3” for Reviewer 9ir2.

3. For **generality across modalities**, please refer to “A3” for Reviewer 9ir2.

---

**Concern 3: Ablation study** -- From Reviewer gkRF, m3Lj, 9ir2

Reviewer gkRF in W2 suggested The Ablation study (Table 3) omits **component-wise impact**.

Reviewers m3Lj and 9ir2 raised questions about the importance of the **proxy model** in Q3 and W2, respectively.

**Our Response**:

1.Explained the proxy model's role and importance in our framework by clarifying its specific function as a hyperparameter generator (not a representation learner).

2.Addressed the misunderstanding regarding our ablation study by providing clear distinctions between ablation experiments for different components.

3.Provided experimental validation with comprehensive results to strengthen our arguments presented in point 1.

**Reference**: Please refer to “A1” for Reviewer gkRF.

---

Beyond the major concerns mentioned above, we also provided detailed responses to all other individual questions and additional experimental results on **baselines, gradient variance, new metrics** and **partial participation**. The reviewers acknowledged that their concerns have been addressed.

Thank you for your consideration.

Best Regards,

Authors of Paper 6648

---

### Decision · Program_Chairs · 2025-09-17

**Decision:**

Accept (poster)

**Comment:**

The paper introduces a framework named FedRAM for Federated Multi-Task Learning (FL-MTL). It's designed to handle challenges such as heterogenous client data and different learning tasks. The key idea is a three-step strategy: A proxy model acts as a middleman to balance contributions from different clients and tasks. This approach dynamically adjusts two key parameters: the task importance weight and the client aggregation coefficient. Experiments show that FedRAM improves performance and cuts down computational costs significantl compared to existing methods. Reviewers agreed that the framework is straightforward, novel in its application to federated learning, and shows significant performance improvements.

Initially, the reviewers raised several concerns despite finding the paper generally well-written and significant. Common concerns included the clarity of some mathematical notations, the need for more justification for certain design choices, and the sensitivity to hyperparameters (ablation studies).  The most significant point, raised by multiple reviewers, was the limited scope of experiments, which focused only on NLP tasks and a specific models. This raised questions about the method's scalability and general applicability to other domains such computer vision. Reviewers also requested robustness tests, such as how the system handles partial client participation.  The authors' rebuttal addressed most of these issues. The reviewers' unanimously confirm that the authors' thorough response addressed their concerns and recommending acceptance of the paper.